

# Participatory flood vulnerability assessment: a multi-criteria approach

Mariana Madruga de Brito[1], Mariele Evers[1], Adrian Almoradie[1]

[1]Department of Geography, University of Bonn, Bonn, 53115, Germany

*Correspondence to*: Mariana Madruga de Brito (mariana.brito@uni-bonn.de)

**Abstract.** This paper presents a participatory multi-criteria decision-making (MCDM) approach for flood vulnerability assessment while considering the relationships between vulnerability drivers. The applicability of the proposed framework is demonstrated in the municipalities of Lajeado and Estrela, Brazil. The model was co-constructed by 101 experts from governmental organizations, universities, research institutes, NGOs, and private companies. Participatory methods such as

the Delphi survey, focus groups and workshops were applied. A participatory problem structuration, in which the modellers work closely with end-users, was used to establish the structure of the vulnerability index. The preferences of each participant regarding the criteria importance were spatially modelled through the Analytical Hierarchy Process (AHP) and Analytical Network Process (ANP) multi-criteria methods. Experts were also involved at the end of the modelling exercise for validation. The final product is a set of individual and group flood vulnerability maps. Both AHP and ANP proved to be

effective for flood vulnerability assessment; however, ANP is preferred as it considers the relationships between vulnerability drivers. The participatory approach enabled experts to learn from each other and acknowledge different perspectives towards social learning. The findings highlight that to enhance the credibility and deployment of model results, multiple viewpoints should be integrated without forcing consensus.

## 1 Introduction

The management of flood risk calls for a better understanding of vulnerability, as hazards only become disasters if they impact a community or system that is vulnerable to their effects (Reilly, 2009). In other words, the vulnerability of the exposed elements will determine whether the hazard will translate into a disaster (Birkmann et al., 2014). Nevertheless, while the understanding of flood hazard has greatly improved over the last decades, the knowledge of vulnerability remains one of the biggest hurdles in risk analysis and improving its assessment is seen as the "missing link" for enhancing our

understanding of risk  (Jongman et al., 2015; Koks et al., 2015).

In general, vulnerability refers to the physical, social, economic and environmental conditions, which increase the susceptibility of the exposed elements to the impact of hazards (UNISDR, 2009). Since vulnerability is not directly measurable, several methods have been proposed to estimate it, including: damage curves (Merz et al., 2010; Papathoma-köhle, 2016), fragility curves (Ozturk et al., 2015; Tsubaki et al., 2016), and vulnerability indicators (Cutter et al., 2003; Roy



and Blaschke, 2013). Both damage and fragility curves are building type-specific and focus on the physical vulnerability of structures to a certain hazard, neglecting the social vulnerability and coping capacity of the inhabitants (Koks et al., 2015). Nevertheless, the ability of the society to anticipate, cope with, and recover from disasters is equally important to assess floods potential impacts. Consequently, several authors emphasize the need for a holistic understanding of vulnerability by

integrating its different dimensions in an overarching framework through the use of indicators (Birkmann et al., 2013; Fuchs et al., 2011; Godfrey et al., 2015).

Indicator-based methods are transparent and easy to use and understand (L. et al., 2013). Since they do not require detailed data as damage and fragility curves, flood vulnerability indicators have been extensively deployed to assess the social vulnerability (Fekete, 2009; Frigerio and De Amicis, 2016), socioeconomic vulnerability (Kienberger et al., 2009), physical

vulnerability (Godfrey et al., 2015; Kappes et al., 2012) as well as to combine multiple dimensions of vulnerability (Roy and Blaschke, 2013; Vojinovic et al., 2016).

Despite the broad variety of motivation and practice, a number of challenges remain in the development of vulnerability indexes as modellers are faced with multiple legitimate choices, thus introducing subjectivity into the modelling process. Key challenges include: (1) selection of the input criteria; (2) data standardization; (3) determination of criteria importance;

(4) consideration of relationships between them; and (5) results validation  (Beccari, 2016; Müller et al., 2011; Rufat et al., 2015). Typically, the rationale for decisions regarding criteria selection, weighting and aggregation is either unstated or justified based on choices made in previous studies. In several cases, no justification is provided at all and the decisions are restricted to project members (Rufat et al., 2015). Surprisingly, notwithstanding the criteria different levels of importance, the vast majority of vulnerability indexes employ an equal weighting (Tate, 2012). Also, even though the dimensions of

vulnerability have diverse and complex linkages among each other (Fuchs, 2009), the relationships between criteria are often neglected and they are assumed to be independent (Chang and Huang, 2015; Rufat et al., 2015). Thus, considering the relationships between vulnerability drivers, their importance weights, and explicitly showing the rationale for model decisions could benefit the development of vulnerability indexes.

In addition to these issues, the participation of multiple stakeholders in the index construction is usually fragmented and

limited to consultation at specific stages. None of the vulnerability indicators reviewed by de Brito and Evers (2016) systematically promoted an active participation throughout the entire vulnerability modelling process. Critical aspects, such as the selection of the input criteria and data standardization were usually constrained to researchers conducting the study. However, participation and cooperation are key aspects for bridging the gap between modellers and end-users and eventually between science and policy (Barthel et al., 2015; Voinov and Bousquet, 2010). If practitioners are involved in creating an

index that they find useful, it is more likely they will incorporate it into policy decisions (Oulahen et al., 2015). Furthermore, better insights can be gained since knowledge beyond the boundaries of an organization is considered. Therefore, a broader and systematic understanding of the problem can be reached, which, in turn, allows designing more effective vulnerability models (Müller et al., 2012).




To tackle these issues, the development of vulnerability indicators could be aided by the use of multi-criteria decision-making (MCDM) tools in combination with participatory methods. MCDM is an umbrella term to describe a set of techniques that can consider multiple criteria to help individuals explore decisions (Belton and Stewart, 2002). The combination of MCDM with participatory tools provides a promising and structured framework for integrating interdisciplinary knowledge in an effort to bring credibility to vulnerability indicators, participant satisfaction and some degree of mutual learning (Sheppard and Meitner, 2005). It can improve the transparency and analytic rigor of flood vulnerability assessment since the choices of input criteria, data standardization, weighting, and aggregation are explicitly expressed, leading to justifiable decisions and reproducible results.

Considering these challenges, we present a participatory approach for assessing the vulnerability to floods by comparing two MCDM methods: the analytical hierarchy process (AHP) and the analytic network process (ANP). We investigate how MCDM tools can be combined with participatory methods to develop vulnerability maps that will be reflective of the local context and trusted by those involved in policymaking. The goal is not to derive a single solution with the "best" flood vulnerability model; instead, our aim is to propose a framework that promotes transparency and integrates contrasting opinions. The approach responds to many of the identified challenges, and, to the best of our knowledge, represents one of the first attempts to apply such a systematic and participatory approach for vulnerability assessment while considering the interdependence among the criteria.

## 2 Study area

Since vulnerability is site-specific (Cardona and van Aalst, 2012), the municipalities of Lajeado and Estrela (274.79 km²), southern Brazil, were used as a case study. In 2016, the total population was approximately 112,000 and the GDP per capita was about US$12,800, with nearly 20% of households living below the poverty line (IBGE, 2017). The regional climate is humid subtropical (Köppen Cfa) and the precipitation is uniformly distributed throughout the year, without a dry season. Rainfall ranges between 1,400 and 1,800 mm per year, with a maximum 24 hours precipitation of 179 mm in 14th April 2011. The discharge is characterized by abrupt flow variations, which are caused by the dense and radial drainage pattern, high mean slope and low soil permeability (Siqueira et al., 2016). The average discharge of the Taquari-River is 321 m³/s with peaks of 10.300 m³/s (FEPAM, 2010). As a consequence of the torrential regimes of rapid runoff, floods occur almost annually, albeit sometimes twice in a year. Between 1980 and 2016, 32 and 34 flood events were reported in Lajeado and Estrela, respectively (Fig. 1). Currently, at least 8,000 persons live in areas with a flood return period of 2 years in these municipalities (CPRM, 2012, 2013). Due to this high susceptibility, Lajeado and Estrela are considered by the Brazilian Government as a priority for disaster risk reduction (CEMADEN, 2017).





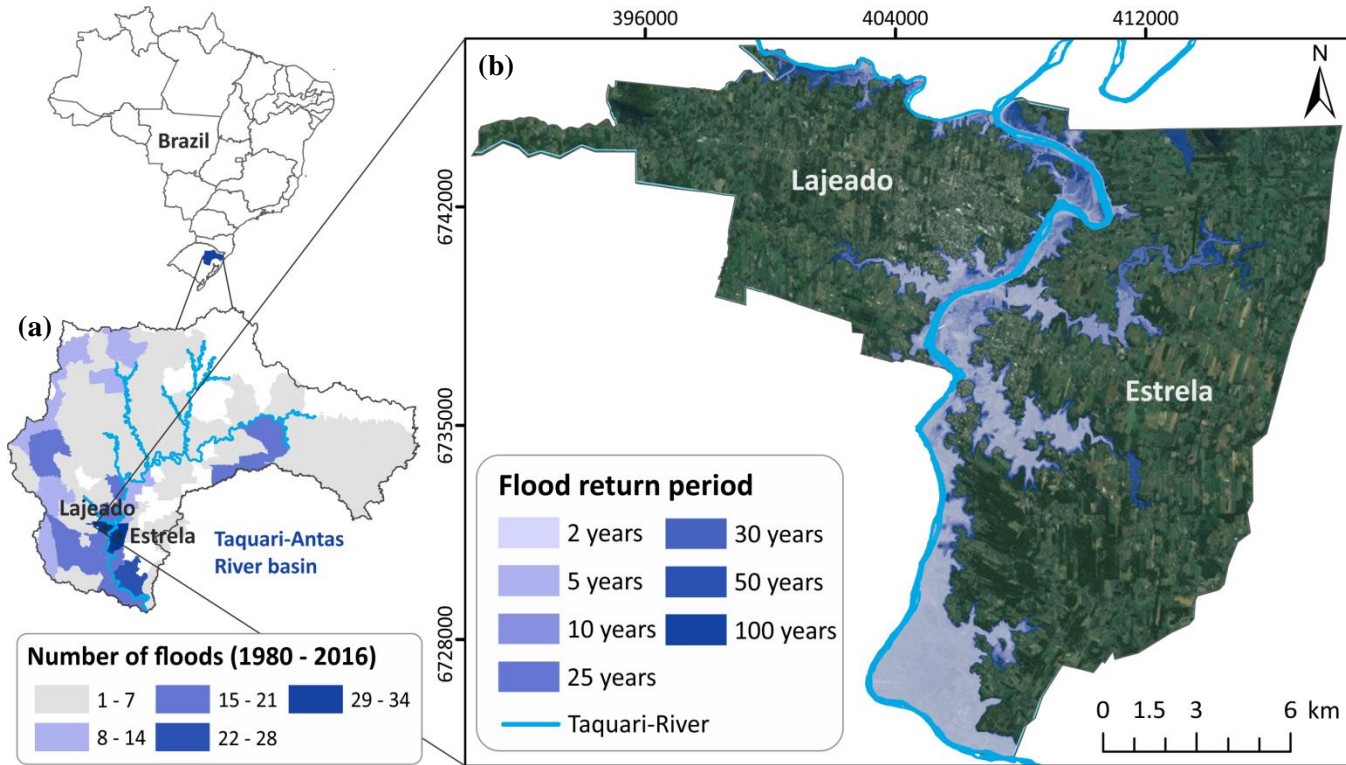

**Figure 1: Location of the study area, southern Brazil: (a) number of floods between 1980 and 2016 in the Taquari-Antas River Basin (elaborated based on Bombassaro and Robaina, 2010; MI, 2017); (b) flood return period in the municipalities of Lajeado and Estrela (Fadel, 2015).**

## 3 Framework for flood vulnerability assessment

The proposed participatory approach for flood vulnerability modelling is summarized in Fig. 2. Experts from governmental organizations, universities, NGOs, and private companies were engaged in all key milestones of the index development. In addition, the partial results of the research were iteratively fed back to participants throughout the entire process to serve as a social learning tool. Participatory techniques which encourage open dialogue such as focus groups and workshops were used to enable experts to exchange knowledge, understand and acknowledge each other positions. A detailed description of the methodological steps will be provided in the following sections.





**Figure 2: Methodological framework for flood vulnerability assessment**

### 3.1 Identification of relevant experts

In this study, we consider an expert as anyone with an in-depth knowledge of flood vulnerability analysis, acquired through

5  experience or education (Krueger et al., 2012). Based on the snowball sampling technique (Wright and Stein, 2005), 117



Brazilian experts were selected. The actors who were cited by more persons were invited to take part in workshops and focus groups in further steps of the study as they play a central role in terms of their reputation and connectedness. A social network analysis depicting the linkages between the selected experts is provided by de Brito et al. (2017).

## 3.2 Selection of vulnerability criteria using the Delphi technique

A two-round Delphi survey was employed to select the input criteria in a systematic and transparent way. The Delphi technique is a structured process for collecting knowledge from a panel of experts using a series of questionnaires interspersed by controlled feedback, seeking to obtain an agreement among the anonymous participants (Linstone and Turoff, 2002). A detailed description of the methods used to prioritize the vulnerability criteria as well as discussion of the results obtained can be found in de Brito et al. (2017).

Based on the Delphi survey, 11 criteria were selected to be included in the vulnerability index. Consensus among participants was reached on all selected criteria, except monthly income. The response rate was 86.32% (n = 101) and 79.20% (n = 80) in the first and second questionnaire, respectively. A description of participants' background, work affiliation and education level can be found in Supplementary Table S1.

The datasets used to represent the selected criteria were obtained mainly from the Brazilian 2010 Census (IBGE, 2010).
Information on the location of persons with disabilities and health care facilities were retrieved from DATASUS (MS, 2016). In addition, interviews were carried out with local civil defence representatives to obtain information on the location of shelters and disaster prevention institutions as well as the number of evacuation drills and training. All datasets were transformed into 20 m resolution raster files.

## 3.3 Structuration of the flood vulnerability index

To proceed with the application of the MCDM tools, a conceptual model with the relationships between the selected criteria needs to be created. The AHP method requires the decomposition of the decision problem into a hierarchy with sub-indexes (e.g. social, economic, etc.). The ANP, on the other hand, uses a network to represent the interaction between criteria and sub-indexes. The elements in this network can be related in any possible way as ANP can incorporate feedback and interdependence relationships.

In this study, a focus group discussion (Morgan, 2005) with 9 participants was conducted to build the AHP and ANP conceptual models. The focus group participants were chosen based on their degree of connectedness (see de Brito et al. 2017). During the meeting, the research objectives and results of the Delphi survey were briefly presented. Then, participants were asked to individually identify the interactions between criteria and organize them into a hierarchy and a network. By soliciting individual schemes, we aimed to avoid the potential bias of experts' responses being influenced by the opinions of
dominant individuals as well as by the pre-existing relationships between them (Frey and Fontana, 1991). Afterwards, the participants verbally put forward their ideas, and when all agreed with a decision, a moderator recorded those on a whiteboard with the support of flash cards. The use of flash cards, rather than writing directly on the whiteboard, allowed for





the criteria to be moved around. When agreement among experts was not met for a specific decision, they were asked to vote by show of hands. All participants were encouraged to contribute to the discussion, which was conducted with minimal intrusion from the researcher. The discussion lasted approximately four hours.

### 3.4 Criteria standardization

Before aggregating the criterion maps into a GIS environment, they need to be transformed into common units as they are represented by different measurement scales (e.g. meters, density/km², etc.). As the selected criteria do not have a linear behaviour, we used value functions to standardize the data. Value functions, also referred as fuzzy membership functions in the GIS literature (Malczewski and Rinner, 2015), avoid setting hard thresholds by recasting the criterion values into a gradual membership of vulnerability ranging from 0 (low vulnerability) to 1 (high vulnerability).

The value function type (e.g. sigmoidal or j-shaped) and the control points that govern their shape were defined in a focus group with 5 participants. The original criteria maps were printed to provide a visual representation of the criteria spatial distribution as well as their minimum and maximum values. Based on that, participants were asked to determine how each criterion contributes to vulnerability. Similarly to the first focus group, the experts' preferences were recorded on a whiteboard. When participants disagreed on a particular choice, they were asked to vote by hand. The collaborative group

discussion lasted about two hours.

### 3.5 Assigning criteria weights using AHP and ANP

It is widely recognized that vulnerability drivers have different levels of importance (Fekete, 2012; Tate, 2012), but it is difficult to find an acceptable weighting scheme. Indeed, assessing the criteria weights is seemed as a sensitive and controversial step in the development of indexes. According to Oulahen et al. (2015), an unweighted index is still subjective

rather than objective, as it treats all criteria as being equally important. Usually, weights are directly assigned by modellers using implicit judgments. In this study, we opted to use the AHP (analytical hierarchy process) and ANP (analytic network process) multi-criteria methods to elicit experts' preferences about criteria weights. The advantage of using structured techniques refers to transparency and results' reproducibility.

In AHP, a reciprocal pairwise matrix is constructed by comparing the criteria and assigning a relative importance value to its

relation according to a 9 point scale (Table 1). This reduces the problem complexity as only two criteria are compared at a time. Once these comparisons are done, the criteria weights are obtained by the principal eigenvector of the matrix (Saaty, 1980).






**Table 1: Scale of relative importance used to compare criteria in AHP and ANP (Saaty, 1980).**

| Numerical rating | Verbal judgment of preferences |
|:---:|:---|
| 1 | Equal importance |
| 3 | Moderate importance |
| 5 | Strong importance |
| 7 | Very strong importance |
| 9 | Extreme importance |

AHP is conceptually easy to use; however, one of its underlying assumptions is that the evaluation criteria are independent. This is a rather strong assumption, especially in the context of spatial problems where interactions among criteria exist (Malczewski and Rinner, 2015). As a solution, Saaty (1999) proposed the ANP, which represents the problem as a network of criteria, grouped into clusters. This provides a more accurate modelling of complex settings by considering criteria inner and outer dependencies. In ANP, similarly to AHP, pairwise comparisons are used to generate matrices of dependent clusters and criteria. The final weights are obtained by using a supermatrix approach. A detailed description of mathematical foundations of ANP and AHP can be found in Saaty (1980, 1999, 2004).

In this study, the hierarchical and network conceptual models were constructed in Super Decisions 2.6.0 software, which automatically created a list with 40 pairwise comparisons needed to run the AHP and ANP evaluations. The AHP comparisons were carried out by asking "which of the two criteria is more important for vulnerability assessment?" while the guiding question in ANP was "which of the two criteria influences a third criterion more with respect to vulnerability assessment?". A questionnaire with these comparisons was prepared in an electronic spreadsheet, and the experts with more connectedness (Brito et al., 2017) were invited to take part in 4 workshops to fill the survey. The workshops started with a presentation of the study objectives, methodology, and preliminary findings. Then, each participant was requested to fill the questionnaire with the 40 comparisons using either the verbal or numeric 9 point scale (Table 2). In the case of the ANP method, the participants could remove any connection between criteria they thought unnecessary. Once the comparisons were done, the weights were automatically displayed in the spreadsheet together with the consistency ratio (CR). The CR measures the probability that the matrix ratings were randomly generated. If the inconsistency was higher than 10%, the experts were asked to revise their judgments. The workshops lasted about 3 hours each and involved a total of 22 participants.

**3.6 Aggregation of criteria to create flood vulnerability maps**

In order to generate the flood vulnerability maps, the standardized criteria were multiplied by the derived weights and subsequently summed. Two scenarios were created for each expert: one with the AHP and the other with the ANP method. In addition, a group scenario was generated by aggregating individual priorities (AIP) using the geometric mean (Ossadnik et



al., 2016). The resultant maps were classified into five categories of vulnerability to facilitate their interpretation and comparison: very low (0.00 – 0.20), low (0.20 – 0.40), medium (0.40 – 0.60), high (0.60 – 0.80), and very high (0.80 – 1.00).

## 3.7 Comparison of AHP and ANP results

The individual AHP and ANP weights were analysed to investigate if the experts' preferences were substantially different from each other and the spatial implications of these differences. The interquartile range (IQR), which is commonly accepted as rigorous way to measure consensus (Giannarou and Zervas, 2014), was used to quantify the degree of conflict between participants regarding the criteria prioritization. The similarities between the individuals were further investigated using cluster analysis with Ward's method (Brusco et al., 2017). In addition, cross tabulation analysis was conducted to compare the spatial distribution of the AHP and ANP vulnerability maps.

## 3.8 Validation

To validate the proposed methodological approach, the opinion of the 22 experts that participated actively in the entire process was considered. Each participant received a report with their own results together with the cluster analysis results. In addition, a Web-GIS platform with the 22 individual and group vulnerability scenarios, flood hazard maps, and historical floods was developed. This allowed participants to have a comprehensive and synthetic view of their results through a customizable user-friendly graphical interface.

Based on the provided feedback, experts were asked about their satisfaction with: (1) the selected criteria; (2) how the criteria were grouped; (3) the weights obtained through the AHP and ANP techniques; (4) the usefulness of the generated vulnerability maps; (5) the feedback received; (6) the transparency of the process; (7) the participatory process as a whole; and (8) the use of the MCDM approach for integrating interdisciplinary knowledge. A 4-point Likert scale was used, ranging from very unsatisfied to very satisfied. Participants were also asked to comment on the difficulty of the MCDM tools and what could be improved in future applications.

## 4. Results

### 4.1 Definition of the structure of the flood vulnerability index

In the first focus group, 9 experts (Supplementary Table S1) co-developed the AHP and ANP conceptual models with the relationships between the selected criteria. A three-level hierarchical tree was built for AHP (Fig. 3a), where the first layer corresponds to the goal, and the second and third levels correspond to the sub-indexes and criteria. Conversely, a network with bilateral relationships was established for the ANP method (Fig. 3b), which enables interactions between criteria situated in different clusters and dependencies between elements in the same cluster to be considered. No fundamental disagreements in the organization of the sub-indexes were evident during the focus group. Nevertheless, minor divergences





occurred in the definition of linkages between criteria on the ANP approach. Despite these challenges, the group succeeded in reaching workable compromises about generic conceptual models that could be used.

The findings of criteria grouping are well-aligned with current guidance on vulnerability (Beccari, 2016; Cardona and van Aalst, 2012), highlighting the importance of coping capacity, as vulnerability is, among other things, the result of a lack of capacity. An emphasis was given to infrastructure aspects which are rarely considered in vulnerability indexes such as the existence of open sewage and accumulated garbage on the street. These criteria play a crucial role in vulnerability assessment in the study area as 54% of the sewage is not piped in Brazil (IBGE, 2011), and the solid waste is commonly disposed on the street in poor neighbourhoods. This causes not only the spread of diseases after floods, but is also a key contributor to localized flooding.

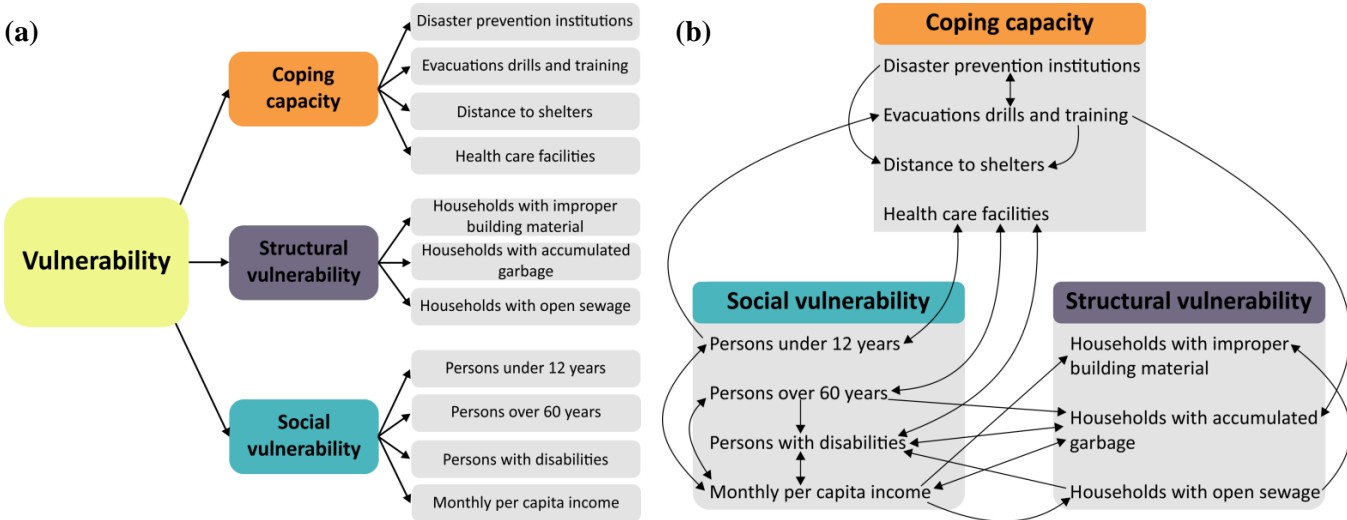

**Figure 3: Conceptual models of the flood vulnerability index: (a) AHP hierarchical tree; (b) ANP network, where the arrow direction indicates the interdependence relationships between criteria. A single direction arrow shows the dominance of one criterion by another. A double direction arrow shows the mutual influence between them**

**4.2 Data standardization**

A shared understanding of the value functions and control points used to standardize the criteria was achieved via a focus group with 5 experts. Due to the small number of participants and since they share a similar background and expertise (Supplementary Table S1), there was an agreement for most decisions taken. Increasing value functions were selected for all social and structural vulnerability criteria, except for the monthly income. Conversely, as a higher coping capacity leads to a reduced vulnerability, decreasing functions were used for coping capacity criteria. Fig. 4 shows the standardized criteria maps, where 0 means low vulnerability and 1 corresponds to high vulnerability.





**Figure 4: Standardized criteria maps, utility functions and control points that govern their shape (a = membership rises above 0; b = membership becomes 1; c = membership becomes 0). The original units used to represent the criteria are shown in parentheses.**





## 4.3 Comparison of AHP and ANP group results

A total of 22 experts attended the workshops designed to fill the AHP and ANP questionnaires (Supplementary Table S1). Overall, the participants had no problems filling the survey. However, due to the large number of pairwise comparisons, some answers needed to be revised as they were contradictory, especially in the AHP technique as the comparison matrices

had more elements.

The group weights derived from the two techniques were similar, except for the monthly per capita income (Table 2). In both methods, the percentage of households with improper building material was the most relevant criterion, closely followed by the number of evacuation drills and other types of training. This importance is partly explained by the high weights attributed to the coping capacity sub-index, which reflects the tendency to widen up the concept of vulnerability to incorporate the

ability of the society to face disasters (Birkmann, 2006), acknowledging that people are not 'helpless victims'.

Agreement among experts about participants weights, measured as an IQR of 20% or less, was achieved only in few variables. In general, the IQR values were lower in the ANP model, indicating higher levels of consensus. The monthly per capita income was the most controversial criterion in the AHP technique and there was a significant divergence among experts about the building material criterion in the ANP model.

**Table 2: Group criteria weights and their respective standard deviation (SD) and interquartile range (IQR). An IQR of 20% or less indicates consensus; 20-30% indicates moderate divergence; 30-40% significant divergence; and >40% strong divergence**

| Sub-index | AHP weight | Criteria | AHP results | | | ANP results | | |
|---|---|---|---|---|---|---|---|---|
| | | | weight | SD | IQR | weight | SD | IQR |
| Social vulnerability | 31.64 | Persons under 12 years | 6.80 | 4.47 | 10.20 | 4.37 | 4.01 | 8.26 |
| | | Persons over 60 years | 6.64 | 4.17 | 17.68 | 3.96 | 2.70 | 6.30 |
| | | Persons with disabilities | 9.39 | 9.97 | 23.03 | 8.84 | 7.51 | 19.30 |
| | | Monthly per capita income | 7.81 | 10.69 | 52.87 | 13.49 | 8.05 | 13.90 |
| Structural vulnerability | 27.7 | Households with improper building material | 14.61 | 9.54 | 34.39 | 15.06 | 10.15 | 28.66 |
| | | Households with accumulated garbage | 6.97 | 7.17 | 28.01 | 7.20 | 7.92 | 23.83 |
| | | Households with open sewage | 7.10 | 9.40 | 22.48 | 6.41 | 7.42 | 20.94 |
| Coping capacity | 40.66 | Disaster prevention institutions | 10.80 | 9.91 | 25.52 | 9.36 | 9.59 | 24.90 |
| | | Evacuation drills and training | 14.17 | 11.87 | 36.79 | 14.54 | 9.98 | 23.96 |
| | | Distance to shelters | 6.42 | 5.23 | 7.32 | 7.26 | 5.56 | 19.64 |
| | | Health care facilities | 9.28 | 7.63 | 19.10 | 9.51 | 7.64 | 14.56 |

The AHP and ANP output maps were compared to identify whether the differences in weights resulted in differences in model outcomes. As shown in Fig. 5, the maps have similar patterns of vulnerability with only minor differences in the

northwest of Lajeado. This difference can be attributed to the lower monthly income in this region. The vulnerability scores from the two models have a linear relationship with a strong correlation ($R^2 = 0.97$) (Fig. 6). Indeed, cross tabulation analysis





showed that 83.11% or 228.39 km² of the study area received the same classification by the two models (diagonal values in Table 3). The main differences were observed in the medium vulnerability class of the AHP model in which 22.73 km² was classified with high vulnerability in the ANP method.

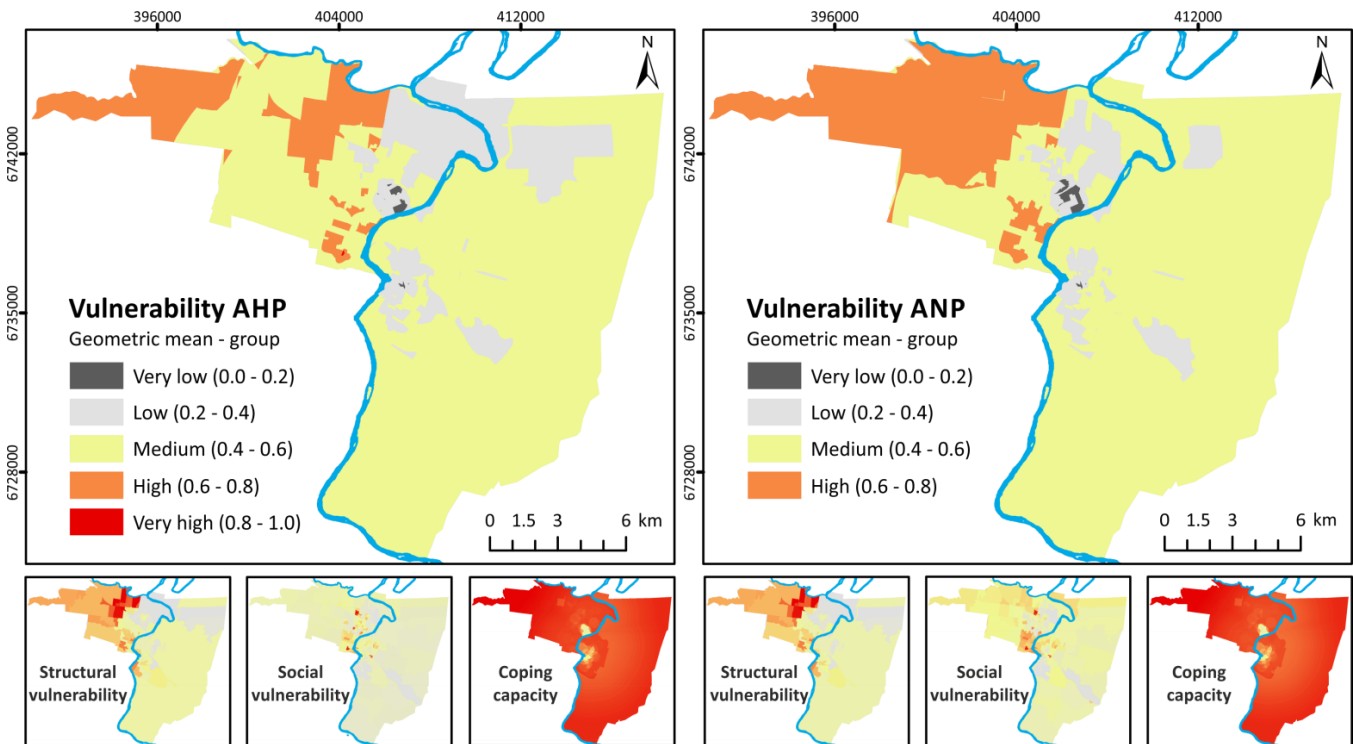

**Figure 5: Spatial distribution of flood vulnerability in the study area.**

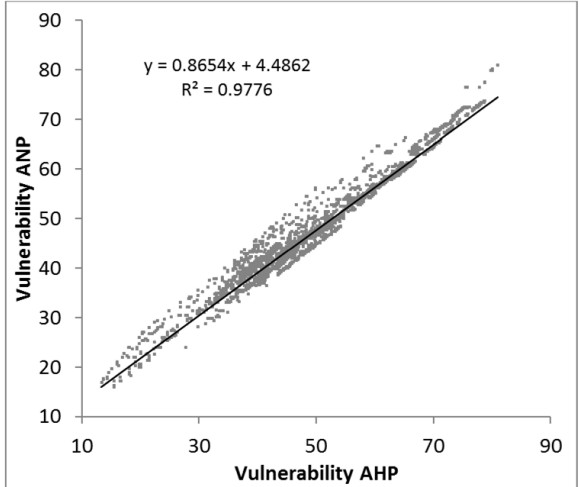

**Figure 6: Correlation of the ANP and AHP flood vulnerability maps scores.**





**Table 3: Comparison of vulnerability classes according to the AHP and ANP models. Diagonal values correspond to areas that were classified equally by both models. The column sum shows the area that is occupied by the respective class of vulnerability in the ANP technique while the line sum shows the area in the AHP technique.**

|  | Vulnerability class | Area ANP (km²) | | | | | Total AHP |
|---|---|---|---|---|---|---|---|
|  |  | Very low | Low | Medium | High | Very high |  |
| **Area AHP (km²)** | Very low | **0.43** |  |  |  |  | 0.43 |
|  | Low | 0.39 | **18.40** | 20.90 |  |  | 39.69 |
|  | Medium |  | 2.25 | **181.82** | 22.73 |  | 206.80 |
|  | High |  |  | 0.13 | **27.74** |  | 27.87 |
|  | Very high |  |  |  | 0.01 | **0.00** | 0.01 |
|  | **Total ANP** | 0.82 | 20.65 | 202.85 | 50.48 | 0.00 | **274.79 km²** |

## 4.4 Comparison of individual weights and scenarios

5   The dispersion of individual weights is illustrated in Fig. 7, where each point represents the weight given to a criterion by one participant. As hinted before by the high IQR and SD values (Table 2), the weights varied significantly across experts, with the greatest differences in the monthly per capita income and households with improper building material items. Given the high degree of disagreement, the aggregation of the individual weights by their mean resulted in loss of information. The points of agreement are criteria that were given a low priority to, such as the percentage of children and elderly.

10   To identify similarities across participants' opinions, we conducted a cluster analysis. The heatmap in Fig. 8 shows the similarities between the participants' priorities. No trends were identified based on their background and work affiliation. Nevertheless, even though individuals hold different points, there is a lot of common ground where importance between criteria is similar. For example, the Architect 2 which works for a government institution has a similar opinion to the Geologist 2, who is a professor.

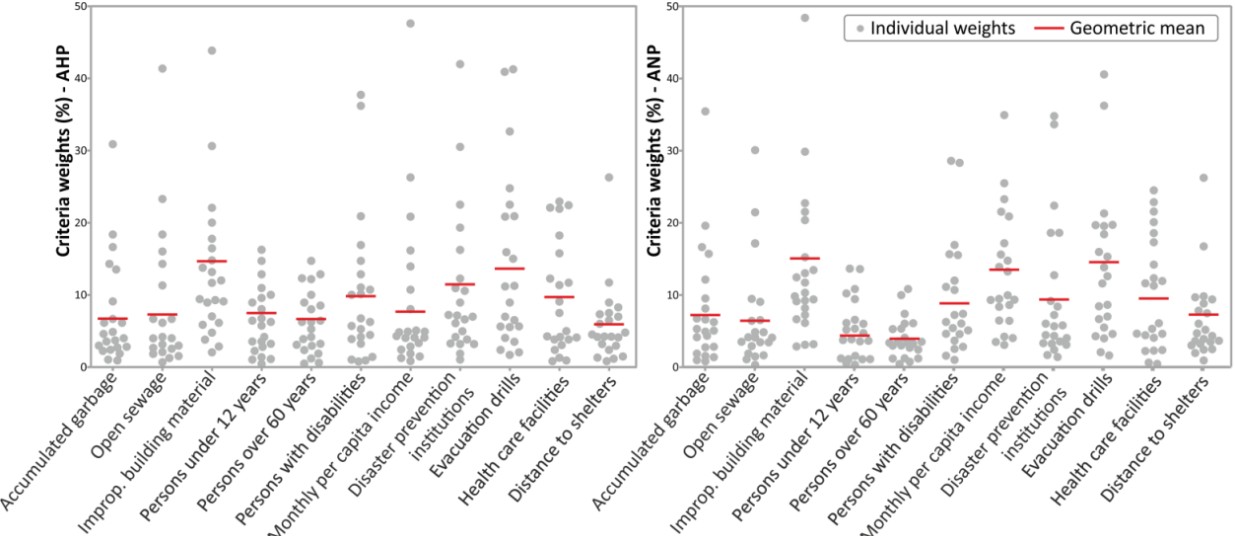

**Figure 7: Diagram of dispersion of individual weight. Each point represents an expert and the red line delineates the mean.**





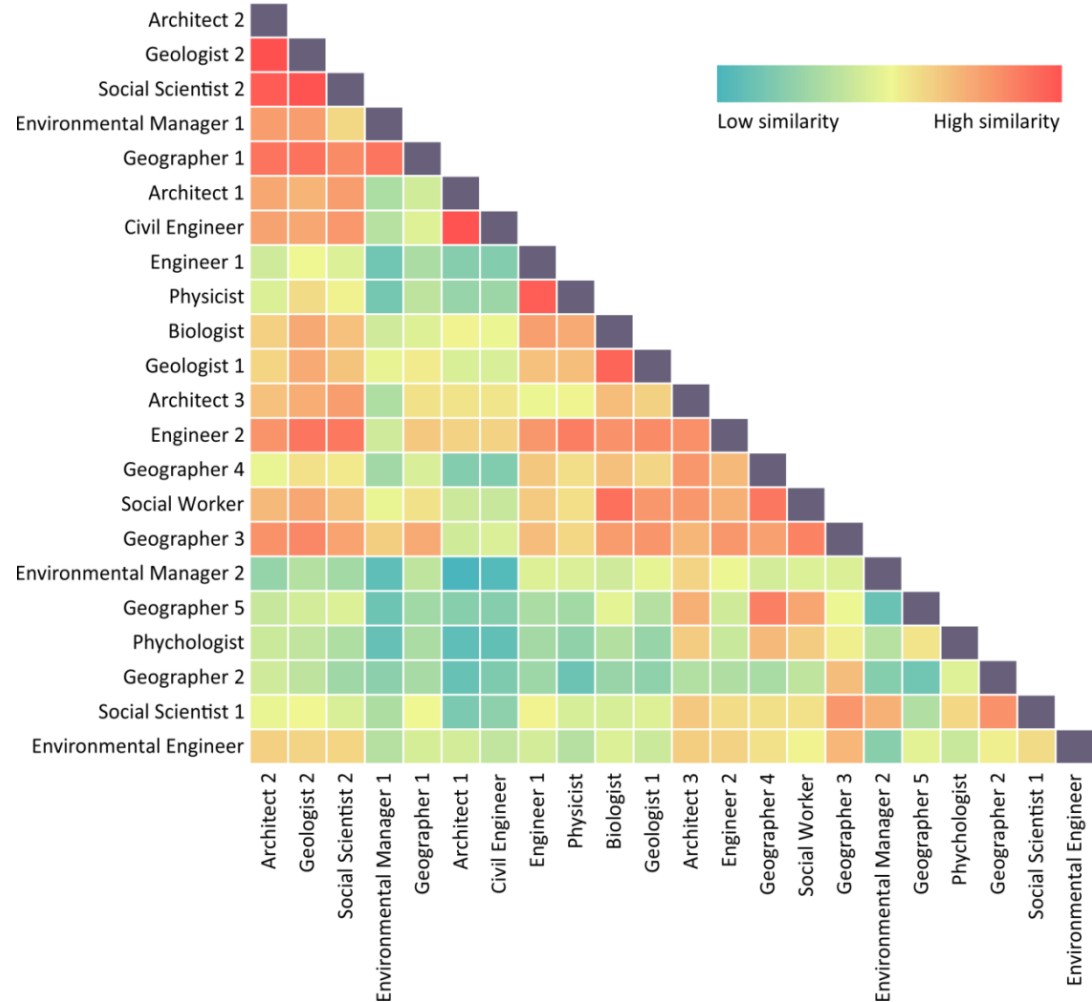

**Figure 8: Heatmap of similarities between experts' weights. The colour gradient from green to red indicates increasing similarity.**

To investigate the spatial implications of the different criteria weights, individual vulnerability scenarios were created for each expert (Supplementary Figure S1). The results demonstrate how different perspectives on criteria weights applied to the

5    same data lead to differences in vulnerability classification. Nevertheless, the trend was similar for both methods, with higher vulnerability values in the northwest of the study area.

A Web GIS platform was set up to allow experts, end-users and the public viewing the model results in form of thematic layers set in a geographical context and overlaid on background data. In this platform (Fig. 9), participants could select their scenarios and compare them with the other participant's results, bringing their positions closer. Also, it was possible to

10    visualize the hazard zones with different return periods, aiming to identify risk areas.



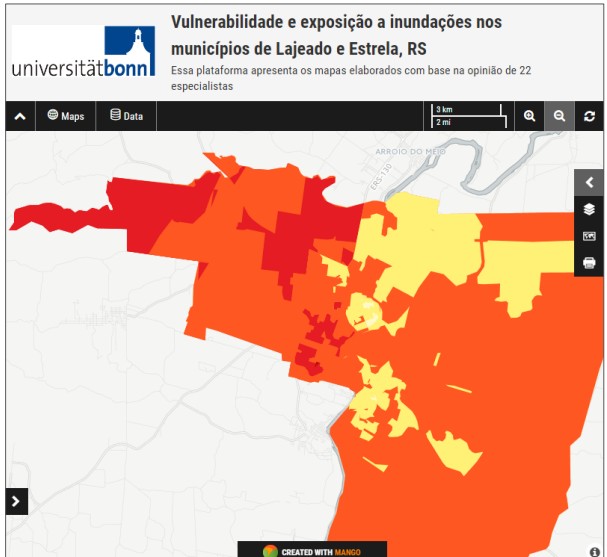

Figure 9: Web GIS platform with the 22 vulnerability scenarios.

## 4.5 Feedback from participants about the proposed participatory MCDM approach

A total of 20 out of 22 invited experts answered the feedback questionnaire. All respondents agreed that the participatory
MCDM approach provides a promising framework for integrating interdisciplinary knowledge in the effort to bring
credibility to vulnerability indexes. Most of them were very satisfied (89%) or satisfied (11%) with the transparency of the
process and with the feedback received. Evaluations of the individual components of the MCDM approach were also
generally positive. All respondents were satisfied or very satisfied with the ANP weights and only one (5%) was unsatisfied
with the AHP results. A total of 50% and 45% of experts were very satisfied and satisfied with the indicators that were
selected, suggesting the Delphi results were representative. Nevertheless, one expert (5%) was unsatisfied with how the
criteria were grouped. Finally, over 53% and 47% respondents indicated that the developed maps are very useful or useful,
respectively. Fig. 10 shows the mean ratings given by participants in each item of the feedback questionnaire.

A number of participants stated that bringing together individuals with different viewpoints resulted in a more
comprehensive and complete view of vulnerability. Quoting a statement from an expert: "the participatory approach allowed
a greater dialogue among stakeholders and encouraged mutual learning, improving the knowledge about multifaceted
problems like flood vulnerability". Several respondents mentioned that the feedback received in form of the web-GIS
platform and partial reports enabled them to see where their response stands in relation to the group. According to them, this
interaction with other experts allowed expanding their knowledge and lead, in some cases, to a change in opinion based on
the information received.

Regarding the difficulty of the MCDM methods used, there was a slight preference for the ANP method. 25% and 20% of
the respondents felt that it was difficult or very difficult to fill the AHP and ANP questionnaires, respectively. In this regard,
one participant stated that the MCDM tools are not applicable to persons with low education levels due to its complexity.





Despite this, experts found easy to grasp the fundamental concepts of AHP and ANP during the workshops, showing enthusiasm about the methodological approach. This was confirmed in the feedback survey, in which the majority showed (85%) interest in applying parts of the proposed method in their future work.

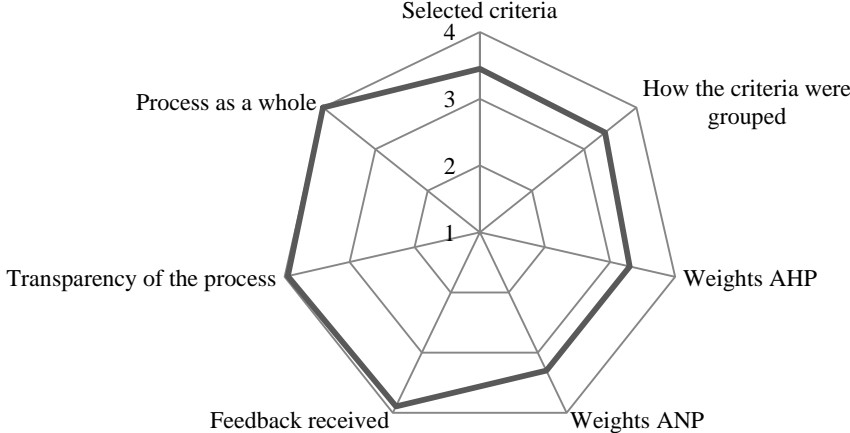

**Figure 10: Participants satisfaction with the participatory process (1 = very unsatisfied; 2 = unsatisfied; 3 = satisfied; 4 = very satisfied).**

## 5 Discussion

### 5.1 Reflections on the participatory process

This study aimed at developing a participatory MCDM approach to assess the vulnerability to floods in an effort to enhance the credibility and deployment of the model outputs. To this purpose, experts were actively involved in all steps of the vulnerability modelling process, thus, having a great influence over the final index. The choices of input criteria, model schematization, data standardization and criteria weighting were done collectively, acknowledging multiple perspectives in a transparent way. By doing so, we avoided that the resulting vulnerability maps were perceived as black boxes by participants since the rationale for key decisions was explicitly expressed, leading to reproducible results. This fostered a sense of ownership among participants which, according to Voinov and Bousquet (2010), brings legitimacy to the model results.

The selection of input criteria using the Delphi technique allowed experts to reframe their personal opinions and reflect on their underlying assumptions through the exchange of information based on the feedback provided and social learning. Further, it gave participants an equal opportunity to contribute without the influence of dominant individuals as all participants remained anonymous. The majority of respondents (95%) were satisfied or very satisfied with the selected criteria, except for one participant. However, as highlighted by Oulahen et al. (2015), the construction of any index is likely to exclude variables considered relevant by some stakeholders.

The two focus groups stimulated in-depth discussions about the structuration of the vulnerability index into sub-indexes and encouraged participants to think about how each criterion contributes to vulnerability in the study area. The elicitation



methods used allowed transforming tacit and implicit knowledge into information useful for vulnerability modelling. Despite some punctual divergences, the participants showed a flexible attitude towards accepting other experts' opinions and succeeded in reaching workable compromises about generic conceptual models and value functions that were satisfactory to all participants. Given the complex nature of the elicitation activities, involvement in the focus groups was restricted to a

limited number of experts to enable them to contribute equally to the discussions. Despite this, the results were representative of the experts' sample as 95% of respondents were satisfied or very satisfied with the developed conceptual models. In this regard, Howarth and Wilson (2006) argue that deliberative processes that are designed to achieve a mutual agreement rather than averaging individual results can enhance the acceptance and quality of the decisions. Nevertheless, it is important to highlight that strong leading voices might have affected the focus group results and overpowered different

opinions.

Overall, the four workshops used to assign the criteria weights worked well as supported by participant's enthusiasm and feedback. The MCDM tools allowed the documentation of different viewpoints about the criteria importance without suppressing dissenting voices, enabling divergent framing assumptions to become explicit. This was central to this study, as vulnerability remains an ill-structured problem (Müller, 2011), where multiple solutions and uncertainty about the input

criteria and their importance exist. Therefore, we believe that systematically showing contrasting views and the underlying reasons for different interpretations is a more transparent approach than deriving a single solution. The aggregation of weights through the geometric mean resulted in loss of information as several prioritizations were reduced to a single vector. Hence, participants whose values are very different from the calculated average may feel that they are not properly represented (Garmendia and Gamboa, 2012). In this regard, van den Hove (2006) argues that forcing consensus by averaging

results in a search for a unique weighting scheme can decrease the legitimacy and effectiveness of participation as a learning process to solve complex problems. Thus, different preferences and conflicts must be recognized and all feasible outcomes should be considered in the decision-making process.

The deliberative feedback throughout the entire process positively impacted the participants' perception of the results transparency, resulting in improved credibility. Consequently, all respondents were very satisfied or satisfied with the

transparency of the methodology and with the feedback received. According to Ledwith and Springett (2009) communication and continuous feedback are essential to the success of any participatory approach as it encourages participants' commitment and interest and may motivate individuals with opposing views to engage in change. In this study, the partial reports, web-GIS platform and final report with cluster analysis results, made explicit potential coalitions, enabling participants to see that they are closer to other professionals than previously perceived.

The validation questionnaire indicated that participants were somewhat likely to agree that the models were clear, trustworthy, and useful, suggesting that participatory modelling activities like the one proposed here are worthwhile. This reinforces the findings of other participatory modelling exercises (Ceccato et al., 2011; Falconi and Palmer, 2017; Kissinger et al., 2017; Maskrey et al., 2016; Oulahen et al., 2015; Voinov and Bousquet, 2010) that state that end-users find it more accurate and useful when the model is created based on their perspectives.



## 5.2 Reflections on the AHP and ANP model results

To analyse the effects of considering the interdependence between criteria in model outputs two MCDM tools were used to elicit experts' preferences about criteria weights. AHP is the most common MCDM method in flood-related studies (de Brito and Evers, 2016). Despite its simplicity, it considers that the criteria are independent of each other, which can be an issue in vulnerability analysis since the magnitude of vulnerability drivers can vary according to inhabitants coping capacity and socioeconomic status (Rufat et al., 2015). For example, the elderly can either be highly vulnerable or less vulnerable depending on their income. To overcome this problem, we used the ANP method, which has a network structure with bilateral relationships, enabling inner and outer dependencies between criteria to be considered (Azizi et al., 2014).

Overall, the criteria weights and ranking were similar in both methods, with the exception of the monthly income. The controversy around the income had already been noticed in the Delphi survey, with this criterion having the lowest degree of consensus among experts. This discrepancy can be explained by the fact that some participants rated it as irrelevant when using the AHP technique. However, when filling the ANP questionnaire, they answered that the income plays a leading role in determining the vulnerability as it influences other criteria such as the building material and households with accumulated garbage or open sewage. Hence, ANP provides a more accurate approach for modelling problems where interrelationships between criteria exist (Saaty, 2004).

Several authors argue that to be accepted and used by stakeholders, models should be simple and easy to use as complexity can obscure transparency and limit model accessibility (Falconi and Palmer, 2017; Horlitz, 2007). During the workshops, it became clear that the elicitation of criteria weights demands a significant cognitive effort from participants due to the inconsistency in the matrices, especially in the AHP technique. Some experts misunderstood the 9 point scale (Table 1) and overused large scores by ranking the criteria they felt more important with 9, regardless of the criteria with which it was being compared. Despite this issue, participants quickly grasped the concepts of the scale and succeed in arriving at consistent judgments. As a result, the majority of them (75% in AHP and 80% in ANP) found it easy or very easy to fill the questionnaires.

The investigation of the spatial implications of the criteria weights showed that the vulnerability scores from the two models are strongly correlated ($R^2 = 0.97$), with 83.11% of the pixels receiving the same classification. Nevertheless, both ANP and AHP models are sensitive to the individual weighting schemes, leading to the creation of different, but equally plausible flood vulnerability maps (Supplementary Figure S1). Even though the general pattern of vulnerability is stable in the study area, a natural question arises given the variability of the vulnerability maps: "which scenario is the best one?" This is still an open question as all scenarios are equally legitimate. As argued by Strager and Rosenberger (2006), MCDM should be used to gain a better insight into the decision-making problem and not as the only or final approach. MCDM makes models more explicit by opening up appraisal inputs to a wider diversity of framings, avoiding simplistic and often misleading one-track solutions (Bellamy et al., 2013; Stirling, 2008).



Experts were, in general, very satisfied with the AHP and ANP results, showing that both methods are effective in solving the ill-structured and interdisciplinary problem of vulnerability. There was a slight preference for the ANP model as participants thought it was easier to understand its logic and no one was unsatisfied with the results. In addition, the agreement among participants about the criteria importance, measured by the standard deviation and IQR, was higher in the

ANP model. Hence, ANP should be adopted whenever possible given that it provides a way to explicit all the relationships among variables. Nevertheless, it should be noted that while AHP can be easily implemented without the need for complex software, ANP requires the use of more sophisticated tools to construct and solve the supermatrix.

### 5.3 Limitations and future research

Although efforts were made to mitigate the risk of bias, some caveats must be acknowledged when interpreting the results

from the developed participatory MCDM approach. Firstly, the small number of participants in the focus groups and workshops poses the risk of unrepresentativeness. This limitation is, according to Garmendia and Stagl (2010), inherent in the nature of participatory modelling processes as they involve normally a small number of participants. Secondly, non-experts were not considered and some groups such as NGOs were underrepresented. Therefore, more case studies as well as the consideration of the opinion of persons who live in flood prone areas are needed to create generalizable conceptual

models to assess the vulnerability to floods in Brazil. Thirdly, some individuals were reticent to express their opinions in the first focus group, requiring encouragement from the moderator. This is a recurrent issue in face-to-face discussions (Anderson and Kilduff, 2009; Then et al., 2014), as the dialogue can be dominated by powerful stakeholders. Hence, the success of focus group relies on the ability of the moderator to keep neutral position, balance participants' involvement and facilitate the consensus building process (Giupponi et al., 2008).

Another criticism is that only a basic approach was used to document the sensitivity of the criteria weights. Further research includes conducting one-at-a-time and global sensitivity analyses to assess the effects of design choices (e.g. standardization, weighting, criteria aggregation) in model outputs. This could be achieved by repeatedly running the model in a Monte Carlo approach (Lilburne and Tarantola, 2009). Alternatively, as global sensitivity analysis is computationally expensive when spatially distributed inputs are considered, simpler approaches such as the procedure described by Chen et al. (2010) could

be used as a starting point. Such analyses would be useful in evaluating the effects of epistemic uncertainty (Walker et al., 2003), helping to understand which choices contribute most to possible variances in the index scores.

Further improvements of the methodology include the conduction of a final workshop to create a vulnerability map by mutual consent. In this setting, the group of participants would determine a weighting scheme that all participants can support. This was suggested by many participants in the feedback questionnaire but was not implemented due to time and

budget constraints. It would also be interesting to carry out a survey at the beginning and the end of the participatory process to investigate how the preferences of participants have evolved over time. This would allow assessing the extent to which social learning occurred. For this purpose, the methods outlined in Garmendia and Gamboa (2012) and Maskrey et al. (2016) could be used.



It is believed that the proposed vulnerability index can be applied to other Brazilian watersheds with similar conditions. However, as it represents the perspective of experts working in Brazil, the findings cannot be generalized to other countries without adaptations. Also, even though the developed approach was applied to flood hazards, the methodology can be used for other types of hazards or even for multi-hazards analysis.

**6 Conclusions**

This study demonstrates how MCDM tools can be used to integrate interdisciplinary knowledge to guarantee not only a useful model according to the needs of the end-users but also to increase the acceptance of the vulnerability maps. The approach proposed herein is particularly novel in the context of vulnerability assessment in the respect that participants actively were involved in all steps of the vulnerability modelling process. This lead to: (1) an increased, shared

understanding of the problem by avoiding the limited perspective of a single expert; (2) an ability to transform implicit and tacit knowledge into information useful for vulnerability modelling; and (3) an enhanced credibility and deployment of the final results.

To the best of our knowledge, this is the first time that the interdependence among criteria was considered to assess the vulnerability to floods. Both AHP and ANP techniques proved to be effective for assessing the vulnerability to floods.

Nevertheless, ANP should be used whenever possible as it allows capturing the complex relationships among vulnerability drivers in a transparent way.

Based on the lessons learned during this participatory process, we can draw some important conclusions. First, if modellers expect the vulnerability model outputs to be used in decision making, end-users should be actively involved in designing it. Second, the search for sound modelling choices should not impose an artificial consensus by averaging individual results.

This is crucial to ensure that the model is legitimized and accepted. Third, MCDM methods which consider interdependence between criteria are preferred for vulnerability assessment given that interrelationships between criteria exist.

From a practical standpoint, the maps created may support local authorities to understand the spatial distribution of vulnerability to floods in the region. The results can also be useful to identify places for site-specific risk assessment, enabling to prioritize human, technological and financial resources and, thereby, improve risk mitigation.

**Acknowledgements**

We are grateful to all experts who participated in the Delphi survey, focus groups and workshops. This work was supported by the Brazilian Coordination for the Improvement of Higher Education Personnel (CAPES) through the grant 13669-13-3.



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
