# Peer review of "Participatory flood vulnerability assessment: a multi-criteria approach"

_Hydrology and Earth System Sciences, 2017_

## Referee Comment (RC1) · Anonymous Referee #1 · 31 Aug 2017

This well-written paper outlines a process for employing a multi-attribute index assessing flood vulnerability. Of particular concern is the participatory approach to constructing this index, allowing for several stakeholders and experts to take part in the construction, essentially when weighting the attributes in terms of their relative importance. The weighting is done using the AHP and ANP approaches, which are popular approaches especially in works employing decision analysis, but sometimes questioned within the decision analysis community itself. Having that said, I wish that the authors addressed other weight elicitation methods within the process outlined in this paper, for an overview see, e.g., the paper M. Riabacke, M. Danielson, L. Ekenberg, "State-of-the-Art Prescriptive Criteria Weight Elicitation," Advances in Decision Sciences, Volume 2012 (2012), Article ID 276584.

[Figure]

For instance, the AHP might restrict the index to fewer attributes than desired due to the requirement for many pairwise comparisons which is avoided in other rank based approaches such as rank-ordered centroid weights or cardinal ranking approaches.

*H. K. Alfares, S. O. Duffuaa, "Assigning cardinal weights in multi-criteria decision making based on ordinal ranking," Journal of Multi-Criteria Decision Analysis, Volume 15, Issue 5-6 , 2008. *M. Danielson, L.ÂăEkenberg, "The CAR Method for Using Preference Strength in Multi-criteria Decision Making," Group Decision and Negotiation, Volume 25, IssueÂă4, 2016.

Further, the text on value functions is a bit hard to penetrate how the value functions were constructed in the participatory process. The selection of only two membership classes for instance (low vulnerability, high vulnerability), was that made by the focus group or designed by the authors on beforehand and why is this sufficient for the model presented?

---

## Referee Comment (RC2) · A. B. Rimba (Referee) · 1 Sep 2017

Dear authors Generally, this research is very interesting for me, you used MCDM approach for flood vulnerability assessment. However, AHP has a complexity which difficult to be understood especially when deciding the criteria and weight of the ranking. This problem was explained in this manuscript. General comment: When I read this research from the beginning (page one) until part 3.8. validation (page 9), I could not imagine what the criteria or parameters for MCDM that used in this research. Difficult to understand the MCDM if we don't know the criteria. In this research, we could find the criteria in the result. Hence, I thought, it will be better and easy to be understood if the authors show and explain the criteria in the method. In some part of the research, the authors referred de Brito (2017). However, I could not find it in the references.

[Figure]

Detail comments: Some determinations are confusing for example; Page 1 line 7; what do you mean vulnerability drivers, please give some explanation or example.

Page 4 figure 1 What do you mean with flood return period? According to Fig.1. dark blue color shows return period 100 years. Does it mean in 100 years, a flood occurs 1 time, and for the soft blue color means in 2 years, flood occur 1 time? If so, 2 years flood return means more vulnerable compared to 100 years. Hence, the color for 2 years should be darker comparing to 100-years flood return.

Page 5 Figure 2 Why is the number of experts different for each criterion?

Page 6 line 10 What are the 11 criteria? Please give the explanation

Page 6 line 18 and 20 How do you transform the data base into 20 m raster resolution? Why do you make it into 20 meters? why don't you select 10 meters or 30meter? What do you mean with "selected criteria" in this manuscript?

Page 9 line 11 What is the opinion of the 22 experts?

Page 9 line 19 Why do you select 4-point Likert scale?

Page Figure 9 What do you mean with 22 vulnerability scenarios?

---

## Author Comment (AC1) · 14 Sep 2017

Dear Referee,

Thanks for reviewing our manuscript. Your comments and observations are very informative and constructive. The manuscript will be revised according to the provided suggestions, which will help to improve the paper before final submission. Please find our response to each one of your comments and questions below.

**1) "This well-written paper outlines a process for employing a multi-attribute index assessing flood vulnerability. Of particular concern is the participatory approach to constructing this index, allowing for several stakeholders and ex-**

[Figure]

perts to take part in the construction, essentially when weighting the attributes in terms of their relative importance."

Thank you for the careful reading and the positive appreciation of our manuscript.

**2) "The weighting is done using the AHP and ANP approaches, which are popular approaches especially in works employing decision analysis, but sometimes questioned within the decision analysis community itself. Having that said, I wish that the authors addressed other weight elicitation methods within the process outlined in this paper, for an overview see, e.g., the paper M. Riabacke, M. Danielson, L. Ekenberg, "State-of-the-Art Prescriptive Criteria Weight Elicitation," Advances in Decision Sciences, Volume 2012 (2012), Article ID 276584.**

**For instance, the AHP might restrict the index to fewer attributes than desired due to the requirement for many pairwise comparisons which is avoided in other rank based approaches such as rank-ordered centroid weights or cardinal ranking approaches.**

**\*H. K. Alfares, S. O. Duffuaa, "Assigning cardinal weights in multi-criteria decision making based on ordinal ranking," Journal of Multi-Criteria Decision Analysis, Volume 15, Issue 5-6 , 2008.**

**\*M. Danielson, L. Ekenberg, "The CAR Method for Using Preference Strength in Multi-criteria Decision Making," Group Decision and Negotiation, Volume 25, Issue 4, 2016.**

Thanks for raising this very important issue. As mentioned by several authors, criteria weighting is one of the most critical and sensitive steps in MCDM (e.g. Edjossan-Sossou et al., 2014; Meyer et al., 2007). Hence, careful attention should be paid to this step when developing vulnerability indexes. In this study, other weighting

methods such as the DEMATEL (decision-making trial and evaluation laboratory) and WINGS (weighted influence non-linear gauge system) were tested with 4 researchers in a previous stage of the research. Nevertheless, they both proved to be too demanding for the respondents.

Therefore, we have opted to use the AHP and ANP methods, which are widely recognized for being easy to use and understand by stakeholders (Dang et al., 2011; Yang et al., 2013). As mentioned by the Referee, AHP is one of the most common MCDM tools in GIS applications (Malczewski and Rinner, 2015) as well as in flood-related studies (de Brito and Evers, 2016). Thus, some of the participants were already familiar with the pairwise comparison approach, making the use of AHP and ANP appealing. The ANP method was chosen because it allows considering the interconnectedness of the input criteria, which was critical to the proposed framework. Indeed, the vulnerability criteria are not independent as multiple interactions between them exist. To overcome this problem, MCDM methods which enable considering inner and outer dependencies between criteria are necessary.

We completely agree with the Referee that a limitation of AHP and ANP is that the number of criteria has to be small. If a large number of criteria is used the procedure becomes operationally too complex and cognitively demanding due to the larger number of pairwise comparisons needed. This was briefly mentioned in the manuscript (see Page 19, lines 17-19). Despite this drawback, the feedback questionnaire showed that most participants are willing to apply the methodology in the future. We will expand the discussion of this limitation in Section 5.3 "Limitations and future research".

An advantage of AHP and ANP when compared to other MCDM methods cited by Riabacke et al. (2012), is that they allow deriving a consistency ratio (CR) of the participants response. The CR offers a useful guide to the validity of the surveyed individual's viewpoint (Mardle et al., 2004). It also allows improving the coherence among redundant judgments (Rahman et al., 2009).

We were not familiar with the SMART, SMARTER, CAR and SWING tools when we conducted the weighting elicitation workshops. To the best of our knowledge, these methods are not commonly applied in the flood risk management community. Indeed, in a systematic literature review of 128 papers conducted in a preliminary step of this study (de Brito and Evers, 2016), only one paper used the SWING approach (Meyer et al., 2009). However, these tools seem to be appropriate for the problem at hand and we agree with the Referee that it would be interesting to use them in future applications.

As mentioned in the paper suggested by the Referee, the centroid weights method provide almost the same accuracy than AHP while requiring much less input and mental effort from decision makers (Alfares and Duffuaa, 2008). Also, in the experiment conducted by Danielson and Ekenberg (2016), the authors found out that CAR and SMART require less amount of time and effort, and are perceived as more transparent than AHP. Therefore, the possibility of applying these methods to weight vulnerability criteria will be added to the discussion section of our manuscript in order to guide future applications.

**3) "The text on value functions is a bit hard to penetrate how the value functions were constructed in the participatory process. The selection of only two membership classes for instance (low vulnerability, high vulnerability), was that made by the focus group or designed by the authors on beforehand and why is this sufficient for the model presented?"**

We did not use classes to build the value functions. Our main intention when using value functions was to avoid defining crisp boundaries. Hence, the data was standardized using a continuous space, ranging from 0 (no vulnerability) to 1 (full vulnerability). The elicitation of the value functions was done with the support of the IDRISI® GIS software.

Due to the complexity of the task, the participation in the focus group was limited to 5 experts. The original criteria maps were printed to provide a visual representation of the criteria spatial distribution as well as their minimum and maximum values. In a first step, the participants had to choose the type of the function used to standardize each criterion. They could be: (1) sigmoidal or s-shaped; (2) j-shaped; (3) linear; or (4) user-defined. The mathematical equations used to build these functions and their basic assumptions are described by Smith et al. (2008).

After selecting the type of the function, the participants had to define if the function was increasing, decreasing or if it was symmetric (Figure 1 below). For example, a decreasing value function was used to standardize the criterion "monthly per capita income" as a higher income leads to a reduced vulnerability. On the other hand, an increasing function was used to standardize the criteria "density of persons under 12 years", since the higher the concentration of children, the more vulnerable the area is.

Finally, the experts had to determine 4 points of the function:

a = membership rises above 0
b = membership becomes 1 (full vulnerability)
c = membership falls below 1
d = membership becomes 0 (no vulnerability)

For example, in standardizing the monthly income criteria, the participants chose to use a sigmoidal decreasing function (Figure 2 below). The control points a, b and c where equal to 510 R$, which was the minimum wage in 2010, when the census data was collected. Hence, the houses with an income of less than 510 R$ were assigned the highest vulnerability scores. The control point d, was set to 2100 R$, which corresponds to the living wage in 2010. In Brazil, the living wage is the minimum income necessary for a worker to meet their basic needs such as shelter, food, transport, health care, clothing, and recreation. We will revise the text and add a detailed description of the procedure adopted to define the value functions.
**References**

Alfares, H. and Duffuaa, S.: Assigning cardinal weights in multi-ÂŘcriteria decision making based on ordinal ranking, J. Multi-Criteria Decis. Anal., 133(March 2008), 125–133, doi:10.1002/mcda, 2008.

Dang, N. M., Babel, M. S. and Luong, H. T.: Evaluation of food risk parameters in the Day river flood diversion area, Red River Delta, Vietnam, Nat. Hazards, 56, 169–194, doi:10.1007/s11069-010-9558-x, 2011.

Danielson, M. and Ekenberg, L.: The CAR Method for Using Preference Strength in Multi-criteria Decision Making, Gr. Decis. Negot., 25(4), 775–797, doi:10.1007/s10726-015-9460-8, 2016.

de Brito, M. M. and Evers, M.: Multi-criteria decision-making for flood risk management: a survey of the current state of the art, Nat. Hazards Earth Syst. Sci., 16(4), 1019–1033, doi:10.5194/nhess-16-1019-2016, 2016.

Eastman, J. R.: IDRISI Kilimanjaro: guide to GIS and image processing, Clark University., 2003.

Edjossan-Sossou, A. M., Deck, O., Al Heib, M. and Verdel, T.: A decision-support methodology for assessing the sustainability of natural risk management strategies in urban areas, Nat. Hazards Earth Syst. Sci., 14, 3207–3230, doi:10.5194/nhess-14-3207-2014, 2014.

Malczewski, J. and Rinner, C.: Multicriteria Decision Analysis in Geographic Information Science., 2015.

Mardle, S., Pascoe, S. and Herrero, I.: Management Objective Importance in Fisheries: An Evaluation Using the Analytic Hierarchy Process (AHP), Environ. Manage., 33(1), 1–11, doi:10.1007/s00267-003-3070-y, 2004.

Meyer, V., Haase, D. and Scheuer, S.: GIS-based multicriteria analysis as decision support in flood risk management., 2007.

Meyer, V., Scheuer, S. and Haase, D.: A multicriteria approach for flood risk mapping exemplified at the Mulde river, Germany, Nat. Hazards, 48(1), 17–39, doi:10.1007/s11069-008-9244-4, 2009.

Rahman, M. R., Shi, Z. H. and Chongfa, C.: Soil erosion hazard evaluation-An integrated use of remote sensing, GIS and statistical approaches with biophysical parameters towards management strategies, Ecol. Modell., 220(13–14), 1724–1734, doi:10.1016/j.ecolmodel.2009.04.004, 2009.

Riabacke, M., Danielson, M. and Ekenberg, L.: State-of-the-art prescriptive criteria weight elicitation, Adv. Decis. Sci., 2012, doi:10.1155/2012/276584, 2012.

Smith, M. J., Goodchild, M. F. and Longley, P. A.: Geospatial analysis: a comprehensive guide to principles, techniques, and software tools., 2008.

Yang, X. L., Ding, J. H. and Hou, H.: Application of a triangular fuzzy AHP approach for flood risk evaluation and response measures analysis, Nat. Hazards, 68(2), 657–674, doi:10.1007/s11069-013-0642-x, 2013.

[Figure]

[Figure]

**Fig. 1.** Types of sigmoidal functions and control points. Source: (Eastman, 2003)

[Figure]

**Fig. 2.** Value function used to standardize the monthly per capita income

[Figure]

---

## Author Comment (AC2) · 14 Sep 2017

Dear Dr. A.B. Rimba,

Thanks for the helpful comments and feedback. The manuscript will be revised according to the provided suggestions, which will help to improve the paper before final submission. Please find our point-by-point response to each one of your comments below.

**1) "This research is very interesting for me, you used MCDM approach for flood vulnerability assessment."**

[Figure]

We are glad that you found the manuscript interesting. Your comments helped to clarify a number of issues

**2) "When I read this research from the beginning (page one) until part 3.8. validation (page 9), I could not imagine what the criteria or parameters for MCDM that used in this research. Difficult to understand the MCDM if we don't know the criteria. In this research, we could find the criteria in the result. Hence, I thought, it will be better and easy to be understood if the authors show and explain the criteria in the method."**

We thank the Referee for having raised this issue. We consider this a good suggestion and that the description of the criteria will improve the quality of the paper. Hence, we will add a table with the selected criteria, their measurement units, and data source in section 3.2 "Selection of vulnerability criteria using the Delphi technique".

**3) "In some part of the research, the authors referred de Brito (2017). However, I could not find it in the references."**

This paper is cited in the reference list (please see Page 22, lines 24-26):

"de Brito, M. M., Evers, M. and Höllermann, B.: Prioritization of flood vulnerability, coping capacity and exposure indicators through the Delphi technique: A case study in Taquari-Antas basin, Brazil, Int. J. Disaster Risk Reduct., 24, 119–128, doi:10.1016/j.ijdrr.2017.05.027, 2017."

We used Mendeley® software using HESS citation style to create the reference list. Even though the name of the first author has a particle "de", the reference was placed under the letter B and not D. We will check with the editorial support personnel of HESS to verify the correct citation format.

**4) "Some determinations are confusing for example; Page 1 line 7; what do you mean vulnerability drivers, please give some explanation or example."**

By vulnerability drivers, we meant the factors that induce and influence vulnerability. In this context, this word could be changed to variables or factors. Although, the term "vulnerability drivers" is commonly used in the scientific literature (e.g. Działek et al., 2016), we will change it to "vulnerability factors" in order to avoid any kind of ambiguity.

**5) "Page 4 figure 1 What do you mean with flood return period? According to Fig.1. dark blue color shows return period 100 years. Does it mean in 100 years, a flood occurs 1 time, and for the soft blue color means in 2 years, flood occur 1 time? If so, 2 years flood return means more vulnerable compared to 100 years. Hence, the color for 2 years should be darker comparing to 100-years flood return."**

Return period is a statistical measure that indicates the average period of time that it takes for a flood to recur at a given location. For example, a flood with a return period of 100 years has a probability of occurrence of 1/100 or 1% in any one year. This does not mean that if a flood with such a return period occurs, then the next will occur in about one hundred years' time. Instead, it means that, in any given year, there is a 1% chance that it will happen, regardless of when the last similar event was (ASCE, 1996).

As the Referee mentioned, the places with a flood return period of 2 years are more susceptible to flooding as there is a higher probability of occurrence when compared with a flood with a return period of 100 years. Thus, we will change the colors of Figure 1 as suggested by the Referee (darker blue for 2-years flood and light blue for 100-years flood).
We did not include a definition of return period in the original text because we consider that the HESS-readers are usually familiar with this concept, since it is common in the hydrology field. Nevertheless, we agree with the Referee that in order to reach a broader public the paper should be understandable by researchers from other fields. Thus, we will add a brief description of return period in the caption of Figure 1.

**6) "Page 5 Figure 2 Why is the number of experts different for each criterion?"**

Thanks for pointing out this issue. The number of experts is not different for each criterion, but for each stage of the index development process. For example, 101 experts participated in the online Delphi Survey. However, it is not possible or desirable to conduct focus groups with such a large group. Thus, as explained in Section 3.1 and fully detailed in a previous paper (de Brito et al., 2017), we conducted a social network analysis to identify key experts. The selected key experts were invited to take part in small workshops and focus groups. The participants of these meetings were different (see Table S1) as the exercises were performed at different localities. We will modify the introductory text of Section 3, as well as the caption of Figure 2 in order to clarify this issue.

**7) "Page 6 line 10 What are the 11 criteria? Please give the explanation"**

The 11 criteria that were selected using the Delphi survey are: (1) persons under 12 years; (2) persons over 60 years; (3) persons with disabilities; (4) monthly per capita income; (5) households with improper building material; (6) households with accumulated garbage; (7) households with open sewage; (8) disaster prevention institutions; (9) evacuation drills and training; (10) distance to shelters; (11) health care facilities. A detailed description of how the criteria were selected is provided by de Brito et al. (2017).

We fully agree that in order to increase the understandability of the text it is better to provide this information in the methods section. Thus, as mentioned in the answer to the comment 2, we will include a Table with the selected criteria, measurement units, and data source in Section 3.2.

**8) "Page 6 line 18 and 20 How do you transform the data base into 20 m raster resolution? Why do you make it into 20 meters? why don't you select 10 meters or 30 meter?**

In order to conduct map algebra more efficiently, all data were transformed to raster files using "polygon to raster" tool in ArcGIS$^{\circledR}$ using the cell center method (ESRI, 2017b). The cell center technique assigns each raster cell the value of the vector object that lies at the geographic centroid of the newly created raster cell. Figure 1 below illustrates how this procedure works in a GIS software.

The choice of 20 meters was not arbitrary. Indeed, it was based on the spatial resolution of the least detailed data used in the analysis, which is the location of health care facilities and location of persons with disabilities. The GPS used to map these locations had an accuracy of 15 – 20 meters. We could have used a larger cell size (e.g., 30 meters) but not smaller, as it is not possible to obtain any more detailed information by resampling the raster to a smaller resolution. We will modify the text to explain that the 20 meters cell size was chosen due to the spatial resolution of the input data.

**9) "What do you mean with "selected criteria" in this manuscript?"**

In this study, the selected criteria are the variables used to assess the vulnerability. The input criteria were selected by 101 experts using the Delphi technique, as explained in section 3.2. We consider that this issue will become clearer when we add

a Table with the results of the Delphi survey (see reply to comment number 2 and 7).

**10) "Page 9 line 11 What is the opinion of the 22 experts?"**

It is the degree of satisfaction of the 22 experts that participated in the AHP and ANP workshops with several aspects of the methodology. In the feedback questionnaire we tried to capture their thoughts about the efficiency of different steps of the proposed methodology (e.g. the AHP weights, the ANP weights, the way the criteria were grouped, the transparency of the process, etc.). We will reformulate this sentence to make it clearer for the readers.

**11) "Page 9 line 19 Why do you select 4-point Likert scale?"**

A 4-point Likert scale was chosen to avoid neutral responses as this scale forces the users to form an opinion (Croasmun and Ostrom, 2011). We will modify the text to explain the choice for a 4-point Likert scale.

**12) "Page Figure 9 What do you mean with 22 vulnerability scenarios?"**

The 22 vulnerability scenarios are shown in Figure S1. They correspond to the individual vulnerability maps created by the 22 experts that participated in the AHP and ANP workshops. In the developed Web-GIS platform, the participants could access all final vulnerability maps created, including the intermediate maps (structural vulnerability, social vulnerability, and coping capacity maps). The caption of Figure 9 will be modified to clarify this issue.

**References**

ASCE: Hydrology handbook, New York., 1996.

Croasmun, J. T. and Ostrom, L.: Using Likert-Type scales in the social sciences, J. Adult Educ., 40(1), 19–22, doi:10.1007/s10640-011-9463-0, 2011.

de Brito, M. M., Evers, M. and Höllermann, B.: Prioritization of flood vulnerability, coping capacity and exposure indicators through the Delphi technique: A case study in Taquari-Antas basin, Brazil, Int. J. Disaster Risk Reduct., 24, 119–128, doi:10.1016/j.ijdrr.2017.05.027, 2017.

Działek, J., Biernacki, W., Fiedeń, Ł., Listwan-Franczak, K. and Franczak, P.: Universal or context-specific social vulnerability drivers – Understanding flood preparedness in southern Poland, Int. J. Disaster Risk Reduct., 19, 212–223, doi:10.1016/j.ijdrr.2016.08.002, 2016.

ESRI: How features are represented in a raster, [online] Available from: http://desktop.arcgis.com/en/arcmap/10.3/manage-data/raster-and-images/how-features-are-represented-in-a-raster.htm (Accessed 5 September 2017a), 2017.

ESRI: How polygon to raster works, [online] Available from: http://pro.arcgis.com/en/pro-app/tool-reference/conversion/how-polygon-to-raster-works.htm (Accessed 5 September 2017b), 2017.

[Figure]

[Figure]

Polygon features

Raster polygon features

**Fig. 1.** Polygon to raster transformation (ESRI, 2017a)

---

## Author Response (AR1)

RHEINISCHE
FRIEDRICH-WILHELMS-
UNIVERSITY OF BONN

DEPARTMENT OF
GEOGRAPHY

UNIVERSITY OF BONN · Department of Geography · Postfach 1147 · D-53001 Bonn

Eco Hydrology and Water
Resources Management

Meckenheimer Allee 166
D-53115 Bonn
phone: +49 (0)228/73-2649
fax: +49 (0)228/73-5607
mariana.brito@uni-bonn.de
www.geographie.uni-
bonn.de/forschung/ags/ag-evers

**RE.: Author's Response - HESS-2017-368**

Dear Editor Dr. Louise Slater,

Bonn, October, 24[th], 2017

Thank you for giving us the opportunity to revise our manuscript. We are thankful to the referees for the thorough review and constructive comments. Based on the given suggestions we have updated the manuscript.

This document contains the author's response. In Section A, the major changes to the original manuscript are briefly summarized. The replies to all comments made by the third reviewer are given in Section B. The authors have answered each comment and explained how the manuscript was modified in light of these suggestions. Finally, Section C contains the marked-up version of the revised manuscript.

We believe that the revised text is significantly improved as a result of the feedback received. Hence, we hope that these changes fulfill the requirements to make the paper acceptable for publication in HESS. If you have any further concerns, please feel free to contact us.

We are very much looking forward to your decision.

Yours sincerely,

Mariana Madruga de Brito and co-authors

**A. List of manuscript changes**

The manuscript was amended in light of the requested revisions. This list aims to highlight the major changes to the original paper.

**1. Introduction**

A definition of the term participatory MCDM was added. Furthermore, to clarify the issues raised by referee 3, we updated the text to make it clear that the aim is not to derive a single vulnerability metric, but to integrate contrasting opinion towards social learning.

**2. Study area**

The colors of the map in Figure 1 were modified as suggested by Dr. A.B. Rimba (referee 2). In addition, a definition of return period was added.

**3. Framework for flood vulnerability assessment**

Following the suggestion of Dr. A.B. Rimba, we created a Table with the selected criteria, their respective data source and metrics used to measure them. As requested by referee 1, the procedure used to define the value functions was further described.

**4. Results**

Some sentences were shortened to improve the text readability. Also, the caption of Figure 9 was modified as suggested by Dr. A.B. Rimba.

**5. Discussion**

This section was profoundly edited to clarify the issues raised by the reviewers. The limitation subsection was rewritten to add the concerns made by referee 1 and 3, including the results representativeness, the complexity of AHP and ANP, and the validation of the developed maps with past flood damages.

**6. Conclusions**

Some word choices were modified to clarify the advantages of the proposed methodology for flood vulnerability assessment.

**B. Point by point response to the comments from referee 3**

Dear Referee,

Thanks for the constructive comments and critics. The manuscript was revised according to the provided suggestions, which we think helped to improve the text. Please find our point-by-point response to each one of your comments below.

**1) This is a stimulating perspective on applying participatory multi-criteria decision-making (MCDM) to vulnerability assessment. However, in the vulnerability assessment field, participatory approaches is commonly used for the general public involvement in the decision making. In the field, what the authors are doing is usually acknowledged as "expert knowledge" or "expert-derived assessment".**

Thank you for this comment, we appreciate that you consider our approach stimulating. We also understand the reviewer's concern regarding the terminology used. However, participatory modeling has many variations regarding the type and level of stakeholder involvement (Robles-Morua et al., 2014; Hedeling et al., 2017). Indeed, the term "participatory" does not necessarily imply that the general public has to be involved.

In the field of MCDM, the terms "participatory" and "participative" are often used to refer to studies which consider the input of key experts (e.g. Arciniegas et al., 2013; Bojorquez-Tapia et al., 2001; Carfora et al., 2016; Derak and Cortina, 2014; Garmendia and Gamboa, 2012; Hossard et al., 2013; Kowalski et al., 2009; Paneque Salgado et al., 2009; Smajgl, 2010). This is also the case of some vulnerability and risk research, which use expert stakeholders knowledge to develop participatory assessments (e.g. Aye et al., 2015; Haase, 2013; Maskrey et al., 2016).

Hence, we consider that the use of the term "participatory" is accurate in the context of our study. We updated the manuscript introduction to clarify what we define as "participatory multi-criteria approach". The following definition was added: "Participatory MCDM refers to a process where a multi-criteria tool is used within participatory settings, where a group of key experts and stakeholders is actively involved (Paneque Salgado et al., 2009)." In addition, two papers that also use the term "participatory MCDM" to involve expert stakeholders were added to the sentence were we first mention this term (page 3, line 2).

**2) Previous studies have produced expert-derived vulnerability assessment using simpler methods, such as scorecard to select, rank and weight the multiple criteria and variables.**

We agree with the referee that previous studies have produced expert-derived vulnerability maps with simpler methods. Still, to the best of our knowledge, and according to a systematic literature review of 128 peer-reviewed papers conducted by de Brito and Evers (2016), we are not aware of any research that incorporated participation in all steps of the vulnerability assessment process to the same extent of our approach. This is highlighted in the conclusion (page 23, lines 3-5).

Usually, the knowledge of experts and key stakeholders is incorporated only in the weights assessment step. Their opinions are not considered in the definition of the input criteria, data standardization or model validation. One of the innovations of our study is that experts were actively involved in all steps of the vulnerability modelling process, and not only to weight the criteria. Thus, participants in our study have a greater sense of ownership and influence over the model results when compared to studies which consult experts only to define weights. This innovation was further explored in the introduction section (see page 2, lines 25-27).

**3) To achieve validation the authors might want to consider the three main issues arising from the literature they mention: - acceptation by the experts involved in the process. Whilst the paper is doing that, it is surprising to collect the feedback of only 22 from the initial 101 experts; it might be seen as some form of retrofitting.**

We thank the referee for having raised this issue. Twenty-two experts were actively involved in the entire modeling process given that it was not realistic to involve 101 persons along the whole process due to time and budget constraints. The remaining 79 experts participated only in the Delphi online questionnaires and, thus, could not evaluate the quality or efficiency of other steps of the index development process. The text was modified to clarify this issue (page 7, lines 7-9; page 9, lines 24-25).

**4) Relying on only 9 expert to build the index and only 5 to calibrate the utility functions might not be seen as the best choice and the authors are right to question the sensitivity of their model.**

This is an excellent point that was hinted in at the discussion. We completely agree with the referee that the sample is small as already acknowledged in the discussion section. The experts selected through snowball sampling are located in different regions of Brazil. Hence, due to resource limitation and time constraints it was not possible to bring them together at one place. This limitation is, according to Garmendia and Stagl (2010), inherent in the nature of participatory modelling processes as they involve normally a small number of participants.

In order to reach a broader number of experts, it would be necessary to use online tools. In an initial step of the project, we considered developing individual utility functions by using web questionnaires. Nevertheless, this would also present a number of drawbacks since the participants would not able to share and hear different perspectives through open dialogue, which is essential for achieving common agreement. In this sense, Mendoza and Martins (2006) argue that group elicitation methods involving open discussion offer several advantages, including the consistency of the information obtained, and a better definition of the preferences. On the other hand, focus group discussions restrict the number of participants to a small sample. By definition, the number of participants in focus groups is small, with between 4 to 12 persons (Carlsen and Glenton, 2011). With more than 12 members, the group becomes difficult to manage and may disintegrate into smaller groups, each having their own independent discussion (Tynan and Drayton, 1988). This was not desired as we wanted to have only one set of utility functions defined by mutual agreement. We added part of this discussion to the limitation section (see page 21, lines 19-27).

Even though only 9 experts were involved in the index structuring phase, 19 out of 20 respondents were satisfied or very satisfied with how the criteria were grouped. This shows that the results were representative of the experts' sample. Therefore, we consider that our approach represents an enhancement in terms of transparency and acceptance of the results when compared to traditional vulnerability studies, where only the knowledge of the modelers (i.e., authors) is considered. Furthermore, we are not aware of any other paper where the input of key experts and stakeholders was considered to standardize the criteria in vulnerability assessment. This step is usually restricted to modelers and analysts. This problem is observed not only in the vulnerability assessment field, but also in MCDM applications in general (Estévez and Gelcich, 2015).

**5) An even better evidence would be actual adoption, ie the experts having enough confidence in the resulting model to actually use it in their decision making. See for example Beccari, B. (2016). A comparative analysis of disaster risk, vulnerability and resilience composite indicators. PLoS currents, 8, https://dx.doi.org/10.1371%2Fcurrents.dis.453df025e34b682e9737f95070f9b970**

We agree with the referee that this would be an ideal validation. In the paper wrote by Beccari (2016), the author mentions that to validate the vulnerability models, the opinion of experts or community members should be considered through the use of surveys, which is exactly what we did.

In the feedback questionnaire, which we sent to the experts, the participants had to answer how useful the results were for their professional activities (see page 9, line 31). All respondents answered that the results are very useful or useful for their work (page 17, lines 12-13). Although this does not mean that the maps are being used in reality, it indicates that they consider using the results. This finding becomes even

more relevant when considering that several respondents work for the local Civil Defense of Lajeado and Estrela and the National Center for Monitoring and Early Warning of Natural Disasters (CEMADEN). Thus, they exert a great influence over decisions related to flood risk management in the region. The text was modified to clarify this issue (see page 19, lines 29-34). Furthermore, Figure 10 was updated to include this information.

**6) Better performance than simpler models. The more is not always the better, and simpler processes might be more cost effective, which is specially important in emerging countries, such as in this case study. The authors might want to confront their results against non-expert derived indexes (the more common are the SOVI and SVI) as well as against simpler expert-derived assessments such as scorecards in order to demonstrate their marginal improvement. See for example Emrich, C. T. (2005). Social vulnerability in US metropolitan areas: improvements in hazard vulnerability assessment. PhD diss., University of South Carolina.**

We do not understand this suggestion entirely because we do not mention that our approach has a "better performance than simpler models" in any part of the manuscript. On the contrary, we criticize authors who believe that their vulnerability models provide "best" results (page 3, line 15). Vulnerability is an ill-structured problem (i.e. a problem for which there is no unique, identifiable, objectively optimal solution) (Rashed and Weeks, 2010). Therefore, vulnerability assessment lacks a single solution algorithm and, in many cases, experts disagree regarding whether a particular choice is appropriate because it has various solution paths (Hong, 1998). This is one of the reasons why we chose to apply multi-criteria methods since the aim of MCDM is not to find a final solution (Kowalski et al., 2009; Roy, 1985), but to deliver a set of alternatives to better inform decision-makers by making subjective judgments explicit in a transparent and fair way.

What we claim is that the results of participatory modelling approaches have a better acceptance when compared to studies conducted without any kind of consultation, participation or collaboration (see modified text page 23, lines 7-8). In addition, they might support social learning processes and develop capacity through awareness raising (Evers et al., 2016). Several papers provide evidence that supports these statements (Hossard et al., 2013; Howarth and Wilson, 2006; Nordström et al., 2010). Hence, we consider that non-expert derived indexes such as the SOVI and SVI will not yield similar results in terms of model acceptance and transparency. We modified the introduction and conclusion sections to clarify this misunderstanding.

We agree with the Referee that, in several cases, simpler processes might be more cost-effective. Intensive participation is cost-intensive and time-consuming. Thus, decisions regarding the degree of participation in certain stages of the modelling process need to be based on a proper balance between conducting a time-efficient process and ensuring that results are representative of local conditions, and trusted

by stakeholders (Andersson et al., 2008). In other words, trade-offs have to be made between the available resources and the expected quality of the MCDM outcomes. We added this information to the section 5.1: "Reflections on the participatory process" (page 20, lines 3-9).

**7) Actual disaster outcomes are usually the main target for validation in the vulnerability assessment field. The authors might want to confront their results against actual outcomes data for their case study or other nearby Brazilian municipalities. Otherwise, their assessment is not robust enough for decision making: it's just asserting and rewarding the judgments that the experts and decision makers already have in that area.**

**See for example Bakkensen, L. A., Fox-Lent, C., Read, L. K., & Linkov, I. (2017). Validating resilience and vulnerability indices in the context of natural disasters. Risk analysis, 37(5), 982-1004;**

**Burton CG (2010). Social Vulnerability and Hurricane Impact Modeling. Natural Hazards, 11 (2): 58-68.**

Thanks for these very interesting papers, which are now cited in our manuscript. We agree with the referee that validation with disaster outcomes is an important element of model building. Nevertheless, validation in participatory modelling activities tends to have a different meaning, since model results are less important as a product in themselves. In some cases the model may have gleaming flaws in terms of calibration and validation, yet they will still be useful for stakeholders to reach a consensus, understand and define their preferences (Voinov and Bousquet, 2010).

It is generally difficult to validate the results of vulnerability analysis as recognized by several authors (Beccari, 2016; Fekete, 2009, 2012). Indeed, in a review of 106 composite indicators, Beccari (2016) found out that only 3 vulnerability models were validated against disaster output information. According to Fekete (2009), one of the main difficulties for validating vulnerability maps is the availability of independent second data source. Even when data is available, the direct comparison of the damage from historical events with the present risk and vulnerability situation is problematic, because in between the two dates there may have been large changes in the land-use (Chen et al., 2016). For example, buildings that were destroyed in the past by floods, might not have been rebuild in the same location, and in other areas new building might have been constructed.

In the study area, data on flood damages is scarce. Unfortunately, the local Civil Defense departments do not collect information on the streets affected by floods, but only at the neighborhood scale. Furthermore, although the number of persons affected is recorded, their spatial location is unknown. The lack of a systematic record of natural disasters damages by the governmental agencies in Brazil make it difficult, if not unrealistic, to perform validation based on actual disaster outcomes. Moreover, we cannot compare or validate it with examples from other areas, as this

would require a similar exercise including hazard modelling, elements-at-risk and vulnerability assessment.

In agreement with the Referee's comment, and since we consider that the limitations should be extensively documented and discussed, we added a paragraph to the discussion explaining this drawback (page 22, lines 6-12).

The validation of the generated maps as well as a sensitivity and uncertainty analysis of the criteria weights will be explored in another paper. As part of the Ph.D. project of the first author, risk maps were generated by combining the vulnerability maps with exposure and hazard maps. Currently, damage data is being collected to validate the final risk maps, which, given the extensive amount of data and work, will be the subject of another paper.

**8) Last but not least, the temptation to merge the multidimensionality of vulnerability into a single metric is prevalent but increasingly criticized in the field. There is a frustrating large gap between the contextual complexity collected among experts and decision makers and the resulting unidimensional correlated metrics.**
**See for example Barnett, J., S. Lambert, and I. Fry. (2008). The hazards of indicators: Insights from the environmental vulnerability index. Annals of the Association of American Geographers 98 (1), 102-19;**
**Rufat, S., Tate, E., Burton, C. G., & Maroof, A. S. (2015). Social vulnerability to floods: review of case studies and implications for measurement, International Journal of Disaster Risk Reduction, 14, 470-486.**

We agree with the referee that it is problematic to merge multiple dimensions of vulnerability in a single value. According to Birkmann (2006) it is difficult – and perhaps even impossible - to reduce the concept of vulnerability to a single equation However, even more problematic is to ignore the different aspects of vulnerability such as the coping capacity, which is often done in vulnerability studies.

Our aim was not to derive a "single metric". As mentioned in the introduction (Page 2, lines 13-14) and throughout the text, the goal was not to provide a single solution with the "best" flood vulnerability model; instead, our aim was to propose a framework that promotes transparency and integrates contrasting opinions towards social learning. This is the main essence of this manuscript: showing the plurality of views by opening up appraisal inputs to a wider diversity of framings (Stirling, 2008). To this purpose, individual vulnerability scenarios were created for each one of the experts, aiming to avoid simplistic and often misleading one-track solutions. As argued by Strager and Rosenberger (2006), MCDM should be used to gain a better insight into the decision-making problem and not as the only or final solution.

We agree with the referee that this is a relevant topic. However, considering that the limitation section is already long and that adding one paragraph to the text would not do justice to the importance of this issue, we opted to not include it in the

discussion section. Moreover, the problem of integrating the dimensions of vulnerability in a single index is discussed in detail in several publications (Prior et al., 2017; UNISDR, 2005). Nevertheless, if the editor and referee think it is necessary we can add a paragraph with this problem.

**References cited in the response**

Andersson, L., Olsson, J. A., Arheimer, B. and Jonsson, A.: Use of participatory scenario modelling as platforms in stakeholder dialogues, Water SA, 34(4 SPEC. ISS.), 439–447, 2008.

Arciniegas, G., Janssen, R. and Rietveld, P.: Effectiveness of collaborative map-based decision support tools: Results of an experiment, Environ. Model. Softw., 39, 159–175, doi:10.1016/j.envsoft.2012.02.021, 2013.

Aye, Z., Jaboyedoff, M., Derron, M.-H. and van Westen, C.: Prototype of a Web-based Participative Decision Support Platform in Natural Hazards and Risk Management, ISPRS Int. J. Geo-Information, 4(3), 1201–1224, doi:10.3390/ijgi4031201, 2015.

Beccari, B.: A Comparative Analysis of Disaster Risk, Vulnerability and Resilience Composite Indicators, PLoS Curr. Disasters, 14(1), doi:10.1371/currents.dis.453df025e34b682e9737f95070f9b970, 2016.

Birkmann, J.: Measuring Vulnerability to Natural Hazards: towards disaster resilient societies, United Nations University Press, Hong Kong., 2006.

Bojorquez-Tapia, L. A., Diaz-Mondragon, S. and Ezcurra, E.: GIS-based approach for participatory decision making and land suitability assessment, Int. J. Geogr. Inf. Sci., 15(2), 129–151, doi:10.1080/13658810010005534, 2001.

Carfora, D., Gironimo, G. Di, Esposito, G., Huhtala, K., Määttä, T., Mäkinen, H., Miccichè, G. and Mozzillo, R.: Multicriteria selection in concept design of a divertor remote maintenance port in the EU DEMO reactor using an AHP participative approach, Fusion Eng. Des., 112, 324–331, doi:10.1016/j.fusengdes.2016.08.023, 2016.

Carlsen, B. and Glenton, C.: What about N? A methodological study of sample-size reporting in focus group studies, BMC Med. Res. Methodol., 11(1), 26, doi:10.1186/1471-2288-11-26, 2011.

Chen, L., van Westen, C. J., Hussin, H., Ciurean, R. L., Turkington, T., Chavarro-Rincon, D. and Shrestha, D. P.: Integrating expert opinion with modelling for quantitative multi-hazard risk assessment in the Eastern Italian Alps, Geomorphology, 273, 150–167, doi:10.1016/j.geomorph.2016.07.041, 2016.

de Brito, M. M. and Evers, M.: Multi-criteria decision-making for flood risk management: a survey of the current state of the art, Nat. Hazards Earth Syst. Sci., 16(4), 1019–1033, doi:10.5194/nhess-16-1019-2016, 2016.

Derak, M. and Cortina, J.: Multi-criteria participative evaluation of Pinus halepensis

plantations in a semiarid area of southeast Spain, Ecol. Indic., 43, 56–68, doi:10.1016/j.ecolind.2014.02.017, 2014.

Estévez, R. A. and Gelcich, S.: Participative multi-criteria decision analysis in marine management and conservation: Research progress and the challenge of integrating value judgments and uncertainty, Mar. Policy, 61, 1–7, doi:10.1016/j.marpol.2015.06.022, 2015.

Evers, M., Jonoski, A., Almoradie, A. and Lange, L.: Collaborative decision making in sustainable flood risk management: A socio-technical approach and tools for participatory governance, Environ. Sci. Policy, 55, 335–344, doi:10.1016/j.envsci.2015.09.009, 2016.

Fekete, A.: Validation of a social vulnerability index in context to river-floods in Germany, Nat. Hazards Earth Syst. Sci., 9(2), 393–403, doi:10.5194/nhess-9-393-2009, 2009.

Fekete, A.: Spatial disaster vulnerability and risk assessments: Challenges in their quality and acceptance, Nat. Hazards, 61(3), 1161–1178, doi:10.1007/s11069-011-9973-7, 2012.

Garmendia, E. and Gamboa, G.: Weighting social preferences in participatory multi-criteria evaluations: A case study on sustainable natural resource management, Ecol. Econ., 84, 110–120, doi:10.1016/j.ecolecon.2012.09.004, 2012.

Garmendia, E. and Stagl, S.: Public participation for sustainability and social learning: Concepts and lessons from three case studies in Europe, Ecol. Econ., 69(8), 1712–1722, doi:10.1016/j.ecolecon.2010.03.027, 2010.

Haase, D.: Participatory modelling of vulnerability and adaptive capacity in flood risk management, Nat. Hazards, 67(1), 77–97, doi:10.1007/s11069-010-9704-5, 2013.

Hedeling, B., Evers, M., Alkan-Olsson, J., Jonsson, A. Participatory modelling for sustainable development: key issues derived from five cases of natural resource and disaster risk management, Environmental Science & Policy, 76, 185-196, doi: 10.1016/j.envsci.2017.07.001, 2017.

Hong, N. S.: he relationship between well-structured and ill-structured problem solving in multimedia simulation, The Pennsylvania State University., 1998.

Hossard, L., Jeuffroy, M. H., Pelzer, E., Pinochet, X. and Souchere, V.: A participatory approach to design spatial scenarios of cropping systems and assess their effects on phoma stem canker management at a regional scale, Environ. Model. Softw., 48, 17–26, doi:10.1016/j.envsoft.2013.05.014, 2013.

Howarth, R. B. and Wilson, M. A.: A theoretical approach to deliberative valuation: aggregation by mutual consent, Land Econ., 82(1), 1–16, doi:10.2307/27647687, 2006.

Kowalski, K., Stagl, S., Madlener, R. and Omann, I.: Sustainable energy futures: Methodological challenges in combining scenarios and participatory multi-criteria

analysis, Eur. J. Oper. Res., 197(3), 1063–1074, doi:10.1016/j.ejor.2007.12.049, 2009.

Maskrey, S. A., Mount, N. J., Thorne, C. R. and Dryden, I.: Participatory modelling for stakeholder involvement in the development of flood risk management intervention options, Environ. Model. Softw., 82, 275–294, doi:10.1016/j.envsoft.2016.04.027, 2016.

Mendoza, G. A. and Martins, H.: Multi-criteria decision analysis in natural resource management: A critical review of methods and new modelling paradigms, For. Ecol. Manage., 230, 1–22, doi:10.1016/j.foreco.2006.03.023, 2006.

Nordström, E.-M., Eriksson, L. O. and Öhman, K.: Integrating multiple criteria decision analysis in participatory forest planning: Experience from a case study in northern Sweden, For. Policy Econ., 12(8), 562–574, doi:10.1016/j.forpol.2010.07.006, 2010.

Paneque Salgado, P., Corral Quintana, S., Guimarães Pereira, Â., del Moral Ituarte, L. and Pedregal Mateos, B.: Participative multi-criteria analysis for the evaluation of water governance alternatives. A case in the Costa del Sol (Málaga), Ecol. Econ., 68(4), 990–1005, doi:10.1016/j.ecolecon.2006.11.008, 2009.

Prior, T., Roth, F., Maduz, L. and Scafetti, F.: Mapping social vulnerability in Switzerland: a pilot study on flooding in Zürich, Zürich., 2017.

Rashed, T. and Weeks, J.: Assessing vulnerability to earthquake hazards through spatial multicriteria analysis of urban areas., 2010.

Robles-Morua, A., Halvorsen, K. E., Mayer, A. S. and Vivoni, E. R.: Exploring the application of participatory modeling approaches in the Sonora River Basin, Mexico, Environ. Model. Softw., 52, 273–282, doi:10.1016/j.envsoft.2013.10.006, 2014.

Roy, B.: Méthodologie Multicritère d'Aide à la Décision, Economica, Paris., 1985.

Smajgl, A.: Challenging beliefs through multi-level participatory modelling in Indonesia, Environ. Model. Softw., 25(11), 1470–1476, doi:10.1016/j.envsoft.2010.04.008, 2010.

Stirling, A.: "Opening up" and "closing down": Power, participation, and pluralism in the social appraisal of technology, Sci. Technol. Human Values, 33(2), 262–294, doi:10.1177/0162243907311265, 2008.

Tynan, A. C. and Drayton, J. L.: Conducting Focus Groups - A Guide for First-Time Users, Mark. Intell. Plan., 6(1), 5, doi:10.1108/eb045757, 1988.

UNISDR: Hyogo Framework for Action 2005-2015: building the resilience of nations and com- munities to disasters, Hyogo., 2005.

Voinov, A. and Bousquet, F.: Modelling with stakeholders, Environ. Model. Softw., 25(11), 1268–1281, doi:10.1016/j.envsoft.2010.03.007, 2010.

**C. Marked-up version of the manuscript**

This section provides the marked-up version of the manuscript. The following symbology was used:

- Text that was inserted appears in red;

- Text that was moved from on section to another appears in green:

- Text that was deleted is shown in balloons;

- Grey vertical track lines in the left margin indicate a change on the adjacent line.

[revised manuscript text omitted]